# The long noncoding RNA lncNB1 promotes tumorigenesis by interacting with ribosomal protein RPL35

Pei Y. Liu [1,19], Andrew E. Tee[1,19], Giorgio Milazzo [2,19], Katherine M. Hannan[3,4], Jesper Maag [5,6], Sujanna Mondal [1], Bernard Atmadibrata[1], Nenad Bartonicek[5,6], Hui Peng[7], Nicholas Ho [1], Chelsea Mayoh[1], Roberto Ciaccio [2], Yuting Sun[1], Michelle J. Henderson[1], Jixuan Gao[1], Celine Everaert [8], Amy J. Hulme [1], Matthew Wong [1], Qing Lan[9], Belamy B. Cheung [1], Leming Shi[10], Jenny Y. Wang [1], Thorsten Simon [11], Matthias Fischer [12], Xu D. Zhang [13], Glenn M. Marshall[1,14], Murray D. Norris [1], Michelle Haber [1], Jo Vandesompele [8], Jinyan Li [7], Pieter Mestdagh [8], Ross D. Hannan [3,4,15,16,17], Marcel E. Dinger [5,18], Giovanni Perini[2]* & Tao Liu [1]*

The majority of patients with neuroblastoma due to *MYCN* oncogene amplification and consequent N-Myc oncoprotein over-expression die of the disease. Here our analyses of RNA sequencing data identify the long noncoding RNA lncNB1 as one of the transcripts most over-expressed in *MYCN*-amplified, compared with *MYCN*-non-amplified, human neuroblastoma cells and also the most over-expressed in neuroblastoma compared with all other cancers. lncNB1 binds to the ribosomal protein RPL35 to enhance E2F1 protein synthesis, leading to *DEPDC1B* gene transcription. The GTPase-activating protein DEPDC1B induces ERK protein phosphorylation and N-Myc protein stabilization. Importantly, lncNB1 knockdown abolishes neuroblastoma cell clonogenic capacity in vitro and leads to neuroblastoma tumor regression in mice, while high levels of lncNB1 and RPL35 in human neuroblastoma tissues predict poor patient prognosis. This study therefore identifies lncNB1 and its binding protein RPL35 as key factors for promoting E2F1 protein synthesis, N-Myc protein stability and N-Myc-driven oncogenesis, and as therapeutic targets.

[1] Children's Cancer Institute Australia for Medical Research, Randwick, NSW 2031, Australia. [2] Department of Pharmacy and Biotechnology, University of Bologna, 40126 Bologna, Italy. [3] Australian Cancer Research Foundation Department of Cancer Biology and Therapeutics, The John Curtin School of Medical Research, The Australian National University, Canberra, ACT, Australia. [4] Department of Biochemistry and Molecular Biology, University of Melbourne, Parkville, VIC 3010, Australia. [5] Garvan Institute of Medical Research, Sydney, Darlinghurst, NSW 2010, Australia. [6] Center for Epigenetics Research, Memorial Sloan Kettering Cancer Center, New York, NY 10065, USA. [7] Advanced Analytics Institute, University of Technology Sydney, Broadway, NSW 2007, Australia. [8] Center for Medical Genetics Ghent, Ghent University, Ghent, Belgium. [9] Department of Neurosurgery, the Second Affiliated Hospital of Soochow University, 215004 Suzhou, Jiangsu, P.R. China. [10] State Key Laboratory of Genetic Engineering, School of Life Sciences and Human Phenome Institute, Fudan University, 201203 Shanghai, China. [11] Department of Pediatric Oncology and Hematology, University Hospital, University of Cologne, Cologne, Germany. [12] Department of Experimental Pediatric Oncology, University Hospital, University of Cologne, Cologne, Germany. [13] School of Medicine and Public Health, Priority Research Centre for Cancer Research, University of Newcastle, Callaghan, NSW 2308, Australia. [14] Kids Cancer Centre, Sydney Children's Hospital, High Street, Randwick, NSW 2031, Australia. [15] Sir Peter MacCallum Department of Oncology, University of Melbourne, Parkville, VIC 3010, Australia. [16] Department of Biochemistry and Molecular Biology, Monash University, Clayton, VIC 3800, Australia. [17] School of Biomedical Sciences, University of Queensland, St Lucia, QLD 4067, Australia. [18] School of Biotechnology and Biomolecular Sciences, University of New South Wales, Sydney, NSW 2052, Australia. [19] These authors contributed equally: Pei Y. Liu, Andrew E. Tee, Giorgio Milazzo. *email: giovanni.perini@unibo.it; tliu@ccia.unsw.edu.au

Cancer is the most common cause of death from disease in children, and neuroblastoma accounts for approximately 15% of all childhood cancer-related deaths. Amplification of the *MYCN* oncogene and consequent over-expression of the N-Myc oncoprotein occur in approximately 25% of human neuroblastoma tissues and correlate with poor patient prognosis[1,2].

N-Myc oncoprotein, like its analog c-Myc oncoprotein, is stabilized when phosphorylated at Serine 62 (S62) by phosphorylated ERK protein[3,4]. Myc oncoproteins induce cell proliferation and tumorigenesis by regulating gene transcription[5,6]. In the last three decades, a number of protein-coding genes have been demonstrated to enhance the tumorigenic effect of N-Myc and c-Myc[5,6]. However, apart from lncUSMycN, CASC15 and NBAT-1[7–9], little is known about the roles of long noncoding RNAs (lncRNAs) in N-Myc-driven neuroblastoma.

LncRNAs regulate gene expression through modulating chromatin architecture, gene transcription[10], precursor messenger RNA splicing[11], mRNA transport, and post-translational modification[12]. Importantly, aberrant lncRNA expression leads to cell proliferation, differentiation block, resistance to apoptosis, chromosome instability, cancer cell migration and invasion, tumor initiation and progression[13–15].

While microarrays have identified a number of protein-coding genes considerably differentially expressed between *MYCN*-amplified and non-amplified human neuroblastoma cell lines, the technology does not cover the majority of lncRNAs. We performed RNA sequencing experiments with RNA from *MYCN*-amplified and non-amplified human neuroblastoma cell lines. Bioinformatics analysis showed that a lncRNA, which we named lncNB1, was one of the most overexpressed transcripts in *MYCN*-amplified neuroblastoma cell lines. lncNB1 binds to the ribosomal protein RPL35, leading to synthesis of E2F1 protein which induces DEPDC1B gene transcription. The GTPase-activating protein DEPDC1B promotes ERK protein phosphorylation, N-Myc protein phosphorylation at S62, and stabilization. The study therefore identifies lncNB1, its binding protein RPL35 and their target DEPDC1B as key factors in N-Myc-driven oncogenesis and as therapeutic targets.

## Results

**LncNB1 is over-expressed in *MYCN*-amplified neuroblastoma.** To identify genes important for N-Myc-induced tumorigenesis, we performed RNA sequencing analysis of four *MYCN*-amplified [CHP134, SK-N-DZ, Kelly, and BE(2)-C] and two *MYCN*-non-amplified (SK-N-AS and SY5Y) human neuroblastoma cell lines. Differential expression analysis revealed 459 genes being differentially expressed between the *MYCN*-amplified and non-amplified samples (Supplementary Data 1).

Consistent with the literature[16–18], *MYCN*-amplified neuroblastoma cell lines showed very high levels of N-Myc, MYCNOS, and IGF2BP1 and very low levels of c-Myc RNA expression (Fig. 1a, b). Interestingly, the lncRNA RP1-40E16.9 at chromosome 6: 3182817–3195767, also known as LOC100507194 and LINC02525 but will herein be referred to as lncNB1 (lncRNA highly expressed in neuroblastoma 1), was polyadenylated and displayed an expression pattern very similar to N-Myc and MYCNOS (Fig. 1a, b), suggesting that this lncRNA could be involved in *MYCN*-amplified neuroblastoma tumorigenesis. The other RNAs considerably over-expressed in *MYCN*-amplified neuroblastoma cell lines were RP11-102F4.3, RNF217, GRIK3, and SLCO5A1 (Fig. 1a, b).

The publicly available Genotype-Tissue Expression (GTEx) Release V7 dataset (dbGaP Accession phs000424.v7.p2) provides RNA sequencing transcript expression data from 53 non-diseased tissues of different sites across nearly 1000 people. Analysis of the GTEx Release V7 dataset showed that lncNB1 was expressed in the brain, pituitary, testis, uterus, and nerve tissues, but hardly detectable in other human tissues (Supplementary Fig. 1a). We next examined lncNB1 expression in human tumor tissues of various organ origins in The Cancer Genome Atlas (TCGA) cohort. As shown in Fig. 1c, lncNB1 was most highly expressed in a proportion of human neuroblastoma tumors, moderately elevated in skin melanoma tissues, but considerably lower in other cancer tissues.

Using the publicly available single-nucleotide polymorphism array data from human neuroblastoma tissues, which were originally generated by the Therapeutically Applicable Research to Generate Effective Treatments (TARGET) initiative[19], we found that the lncNB1 gene was gained in six and deleted in 1 of the 341 neuroblastoma tissue samples (Supplementary Table 1), suggesting that lncNB1 gene copy number change was uncommon in neuroblastoma.

By RT-PCR analysis, we confirmed that lncNB1 was expressed at considerably higher levels in five *MYCN*-amplified, compared with three *MYCN*-non-amplified neuroblastoma cell lines (Fig. 1d). Using the publicly available Maris-41-FPKM-rsg001 RNA sequencing dataset from 39 human neuroblastoma cell lines [http://r2.amc.nl][20], two-sided Pearson's correlation study showed that lncNB1 RNA expression positively correlated with N-Myc mRNA expression (Fig. 1e), and lncNB1 was expressed at a significantly higher level in a subset of human neuroblastoma tumors with *MYCN* amplification (Fig. 1f). In addition, RT-PCR analysis showed that lncNB1 RNA was mainly localized in the cytoplasm not in the nucleus (Supplementary Fig. 1b, c).

Taken together, the data suggest that lncNB1 expression is the highest in neuroblastoma compared with all other human cancers, and correlates with *MYCN* gene amplification and expression.

**LncNB1 up-regulates *DEPDC1B* gene and E2F1 protein expression.** LncRNAs are well-known to regulate the expression of neighboring protein-coding genes[10,21,22]. Our RT-PCR analysis showed that transfection of BE(2)-C and CHP134 neuroblastoma cells, which express high levels of lncNB1 (Fig. 1d), with lncNB1 siRNAs, siRNA-1, or siRNA-2, did not have an effect on the expression of TUBB2A or TUBB2B (Supplementary Fig. 2a, b), the neighboring protein-coding genes of lncNB1.

To examine the effect of lncNB1 in regulating gene expression in trans, BE(2)-C cells were transfected with control siRNA, lncNB1 siRNA-1, or siRNA-2 for 40 h. Affymetrix microarray differential gene expression analysis revealed that knocking down lncNB1 modulated the expression of a number of target genes (Supplementary Table 2), among which DEPDC1B was a potentially important candidate gene as it is known to induce ERK protein phosphorylation[23,24] and phosphorylated ERK is known to enhance N-Myc protein stability[3,4]. Gene set enrichment analysis showed that the most repressed transcription factor-binding sites, after lncNB1 knockdown, were binding sites for E2F1 and its partners E2F1-DP1 and E2F1-DP2 (Fig. 2a, Supplementary Table 3). RT-PCR and immunoblot analyses confirmed that lncNB1 siRNAs consistently reduced DEPDC1B mRNA and protein as well as E2F1 protein, but did not show a consistent effect on E2F1 mRNA, in BE(2)-C, Kelly and CHP134 neuroblastoma cells (Fig. 2b, c).

We next established BE(2)-C and Kelly cells stably expressing doxycycline-inducible control shRNA, lncNB1 shRNA-1 or lncNB1 shRNA-2, which targeted lncNB1 RNA regions different from lncNB1 siRNAs. RT-PCR and immunoblot analyses confirmed that treatment with doxycycline consistently reduced lncNB1 and DEPDC1B mRNA as well as DEPDC1B and E2F1 protein but not E2F1 mRNA expression in doxycycline-inducible

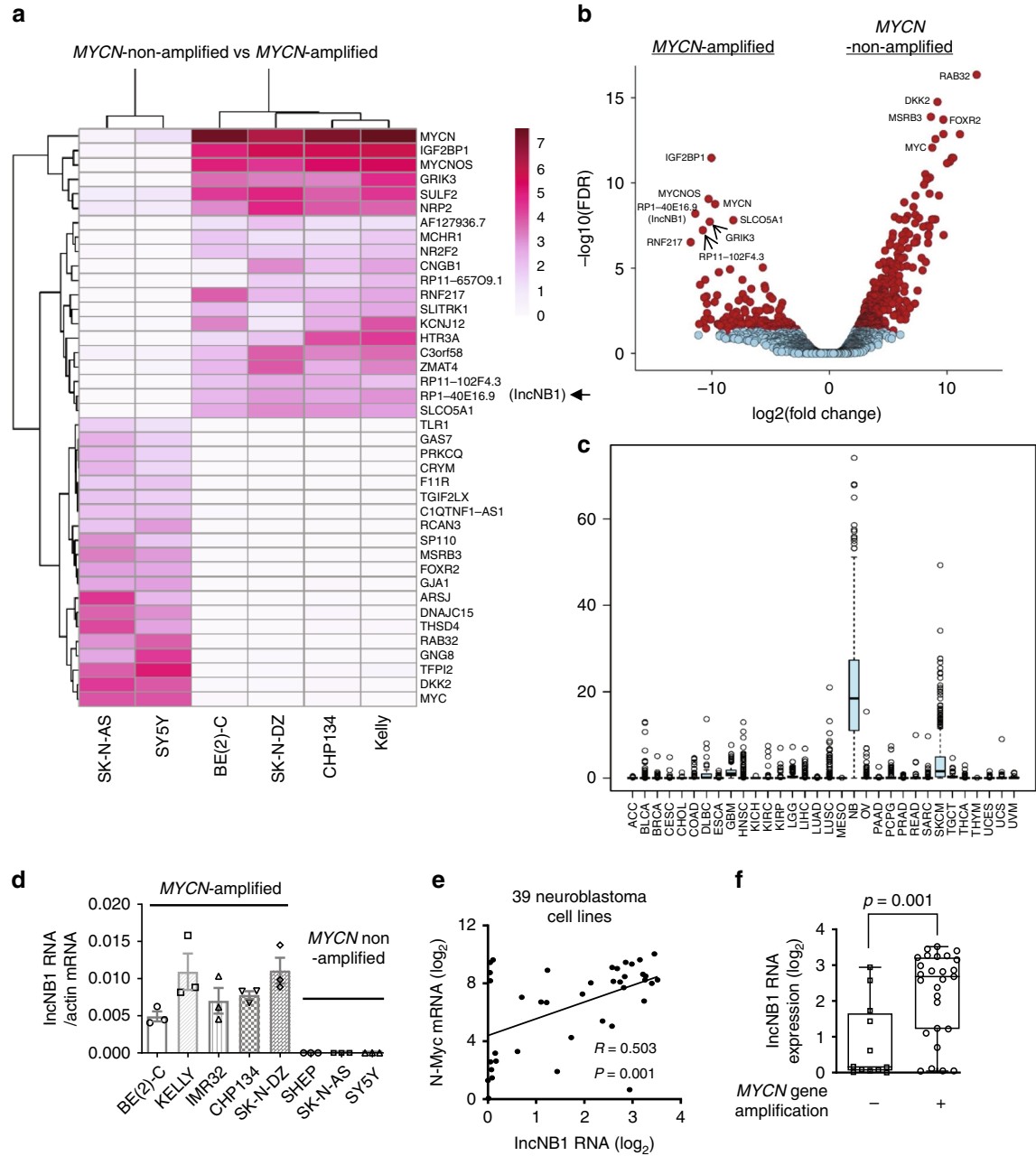

**Fig. 1** LncNB1 is over-expressed in *MYCN*-amplified human neuroblastoma tissues and cells. **a**, **b** RNA was extracted from four *MYCN*-amplified (BE(2)-C, Kelly, CHP134, and SK-N-DZ) and two *MYCN*-non-amplified (SY5Y and SK-N-AS) neuroblastoma cell lines. Heatmap showed the top 40 genes most differentially expressed between the two groups of cell lines (**a**). Volcano plot revealed the top genes most significantly differentially expressed between the cell lines with the lowest false discovery rate (**b**). **c** lncNB1 RNA expression in human tumor tissues of various organ origins in The Cancer Genome Atlas (TCGA) cohort. The center line was the median, the bounds of the box were the upper and lower quartiles and the whiskers extended to 1.5× the inter-quartile range. Abbreviations of cancers were explained in the Methods section, and NB and SKCM represented neuroblastoma and skin melanoma. **d** RNA was extracted from *MYCN*-amplified and *MYCN*-non-amplified human neuroblastoma cell lines, followed by RT-PCR analysis of lncNB1 RNA expression. Data were shown as the mean ± standard error of three independent experiments. **e**, **f** lncNB1 and N-Myc RNA expression was extracted from the publicly available Maris-41-FPKM-rsg001 RNA sequencing dataset from 39 human neuroblastoma cell lines downloaded from R2 microarray analysis and visualization platform [http://r2.amc.nl]. Correlation between lncNB1 and N-Myc RNA expression was analyzed by two-sided Pearson's correlation (**e**). The center line was the median, the bounds of the box were the upper and lower quartiles, and the whiskers were set to the minimum and maximum values ($p = 0.001$, two-sided unpaired Student's *t*-test). Data were shown as the mean ± standard error (**f**). Source data are provided as a Source Data file

lncNB1 shRNA cells (Fig. 2d, e, Supplementary Fig. 2c). Taken together, the data demonstrate that lncNB1 up-regulates *DEPDC1B* gene expression and increases E2F1 protein expression.

**LncNB1 up-regulates N-Myc protein expression through DEPDC1B.** N-Myc protein is stabilized after phosphorylation at

S62 by phosphorylated ERK[3]. Since DEPDC1B is known to induce ERK protein phosphorylation[24], we examined whether lncNB1 regulated ERK protein phosphorylation, N-Myc protein phosphorylation, and stabilization. Transfection with lncNB1 siRNAs did not alter N-Myc mRNA expression (Supplementary Fig. 3a), but considerably reduced DEPDC1B protein expression, ERK

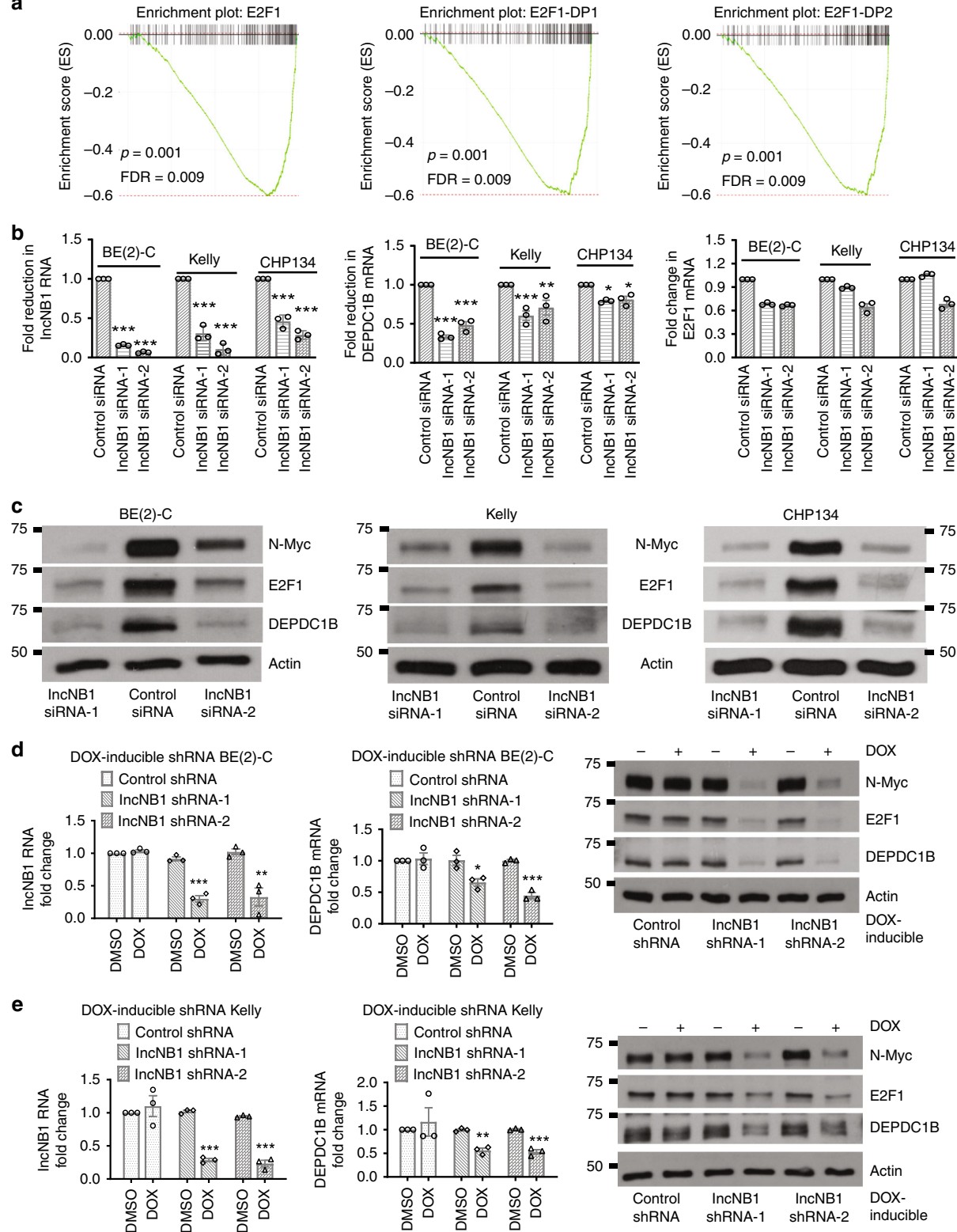

**Fig. 2** LncNB1 is required for DEPDC1B mRNA and protein and E2F1 protein expression. **a** Genome-wide differential gene expression was examined with Affymetrix microarray in BE(2)-C cells 40 h after transfection with control siRNA, lncNB1 siRNA-1, or siRNA-2. Gene set enrichment analysis generated histograms confirming down-regulation of E2F1 target genes by lncNB1 siRNAs. FDR indicated false discovery rate. **b, c** BE(2)-C, Kelly, and CHP134 cells were transfected with control siRNA, lncNB1 siRNA-1, or lncNB1 siRNA-2 for 48 h. LncNB1, DEPDC1B, and E2F1 RNA expression was analyzed by RT-PCR (**b**), and DEPDC1B, E2F1, and N-Myc protein expression was analyzed by immunoblot (**c**). Data were shown as the mean ± standard error of three independent experiments, and evaluated by one-way ANOVA. *, **, and *** indicate $P < 0.05$, 0.01, and 0.001 respectively. **d, e** Doxycycline (DOX)-inducible control shRNA, lncNB1 shRNA-1 or shRNA-2 BE(2)-C (**d**) and Kelly (**e**) cells were treated with vehicle control or DOX for 48 h. RNA and protein were extracted from the cells for RT-PCR and immunoblot analyses of lncNB1, DEPDC1B, E2F1, and N-Myc. Data were shown as the mean ± standard error of three independent experiments, and evaluated by two-tailed unpaired Student's $t$-test. *, **, and *** indicate $P < 0.05$, 0.01, and 0.001 respectively. Source data are provided as a Source Data file

protein phosphorylation, N-Myc protein phosphorylation at S62 and expression in BE(2)-C, Kelly, and CHP134 cells (Fig. 3a). Consistent with these data, transfection with DEPDC1B siRNAs reduced DEPDC1B but not N-Myc mRNA expression (Supplementary Fig. 3b, c), and reduced DEPDC1B protein expression, ERK protein phosphorylation, N-Myc protein phosphorylation at S62 and expression in the three neuroblastoma cell lines (Fig. 3b). In addition, transfection with a lncNB1 or DEPDC1B over-expression construct in BE(2)-C and Kelly cells up-regulated DEPDC1B protein expression, ERK protein phosphorylation as well as N-Myc protein expression (Supplementary Fig. 3d). DOX-inducible lncNB1 shRNA BE(2)-C and Kelly cells were then transfected with an empty vector or DEPDC1B expression construct and treated with vehicle control or DOX for 48 h. Immunoblot analysis showed that forced DEPDC1B over-expression largely reversed lncNB1 shRNA-mediated ERK protein dephosphorylation, N-Myc protein dephosphorylation at S62, and N-Myc protein reduction (Supplementary Fig. 3e).

BE(2)-C cells were then transfected with control siRNA, lncNB1 siRNA-1, or DEPDC1B siRNA-1 for 30 h, followed by treatment with vehicle control, the protein synthesis inhibitor cyclohexamide, or the proteasome inhibitor MG-132. In addition, BE(2)-C cells were transfected with an empty vector or lncNB1 expression construct, followed by treatment with vehicle control or cycloheximide. The pulse chase assays showed that N-Myc protein half-life was reduced by approximately 50% by lncNB1 siRNA or DEPDC1B siRNA (Fig. 3c), and was increased by approximately 39% by the lncNB1 expression construct (Supplementary Fig. 3f). While MG-132 did not increase DEPDC1B protein expression in cells transfected with lncNB1 siRNA or DEPDC1B siRNA, which ablated DEPDC1B mRNA, MG-132 dramatically up-regulated N-Myc protein expression in cells transfected with lncNB1 siRNA or DEPDC1B siRNA (Fig. 3d). The data demonstrate that lncNB1 and DEPDC1B are required for N-Myc protein stability.

Stable doxycycline-inducible control shRNA, lncNB1 shRNA-1 or lncNB1 shRNA-2 BE(2)-C and Kelly cells were then treated with vehicle control or doxycycline for 48 h. Immunoblot analysis confirmed that knocking down lncNB1 reduced DEPDC1B and N-Myc protein expression, ERK protein phosphorylation, and N-Myc protein phosphorylation at S62 (Fig. 3e).

To exclude cell growth inhibition or cell death as a contributing factor in the regulation of N-Myc expression, we transfected BE(2)-C and Kelly cells with control siRNA, lncNB1 siRNAs or DEPDC1B siRNAs for 48 h, and treated DOX-inducible control shRNA and lncNB1 shRNA BE(2)-C and Kelly cells with vehicle control or DOX for 48 h. Alamar blue assays and flow cytometry analyses of Annexin V positively stained cells showed that knocking down lncNB1 or DEPDC1B expression for 48 h was too early to have an effect on neuroblastoma cell proliferation or survival (Supplementary Fig. 4a–e). Taken together, the data show that lncNB1 is required for N-Myc protein expression through DEPDC1B gene transcription, ERK protein phosphorylation, N-Myc protein phosphorylation at S62 and thus N-Myc protein stability.

**LncNB1 activates *DEPDC1B* gene transcription through E2F1.** To further investigate whether lncNB1 can regulate DEPDC1B and E2F1 gene transcription, we performed chromatin immunoprecipitation (ChIP) assays with a control IgG or an antibody against tri-methylated histone H3 lysine 4 (H3K4me3), a marker for active gene transcription, followed by PCR with primers targeting a negative control, the DEPDC1B or the E2F1 gene core promoter region (Supplementary Fig. 5a). The ChIP assays showed that knocking down lncNB1 reduced H3K4me3 at the DEPDC1B, but not the E2F1, gene core promoter (Supplementary

Fig. 5b), further demonstrating that lncNB1 regulates DEPDC1B, but not E2F1, gene transcription.

We then cloned the DEPDC1B gene promoter regions 1146, 545, or 75 bp upstream of the transcription start site into the pGL3 firefly-luciferase construct (Fig. 4a), and transfected the constructs into doxycycline-inducible control shRNA, lncNB1 shRNA-1, or lncNB1 shRNA-2 BE(2)-C cells. Dual luciferase assays showed that knocking down lncNB1 expression with doxycycline considerably reduced luciferase activity of all the wild type constructs (Fig. 4a), suggesting that the 75 bp region upstream of the DEPDC1B transcription start site was essential and sufficient for lncNB1 to activate DEPDC1B gene transcription. As a 20 bp nucleotide section in the 75 bp region was enriched of E2F1-binding sites, we generated DEPDC1B gene promoter pGL3 constructs deleted of the 20 nucleotides (Fig. 4a). Dual luciferase assays showed that deletion of the 20 nucleotides abolished the effect of lncNB1 shRNAs in reducing DEPDC1B promoter activity (Fig. 4a).

Since RT-PCR analysis showed that lncNB1 RNA was mainly localized in the cytoplasm not in the nucleus (Supplementary Fig. 1b, c), we investigated whether lncNB1 induced DEPDC1B gene transcription via E2F1 activity. Knockdown of E2F1 expression with two independent siRNAs reduced DEPDC1B mRNA and protein expression in both BE(2)-C and Kelly neuroblastoma cell lines (Fig. 4b, c), and replicated the effects of lncNB1 siRNAs, lncNB1 shRNAs, and DEPDC1B siRNAs in reducing ERK protein phosphorylation, N-Myc protein phosphorylation at S62 and N-Myc protein expression in BE(2)-C and Kelly cells (Supplementary Fig. 5c). ChIP assays showed that E2F1 protein was highly enriched at the DEPDC1B gene core promoter region close to the transcription start site, and that knocking down lncNB1 expression with doxycycline considerably reduced E2F1 protein binding at the DEPDC1B gene core promoter in doxycycline-inducible lncNB1 shRNA-1 and shRNA-2 BE(2)-C cells (Fig. 4d). Moreover, luciferase assays showed that forced over-expression of E2F1 and its functional partner DP1 significantly enhanced wild-type DEPDC1B gene promoter activity, and that the effect was abolished by deletion of the E2F1-binding site-enriched 20 nucleotides of the DEPDC1B gene promoter (Fig. 4e). Taken together, the data indicate that lncNB1 activates DEPDC1B gene transcription through increasing E2F1 protein expression.

**LncNB1 induces E2F1 protein translation by binding to RPL35.** To identify the mechanism by which lncNB1 increases E2F1 protein but not mRNA expression, we in vitro-transcribed lncNB1 RNA from full-length lncNB1 cDNA products containing the T7 promoter in the sense strand (experimental) or SP6 promoter in the antisense strand (negative control) and labeled the RNA with biotin. The biotin-labeled lncNB1 RNA was then incubated with protein lysates from BE(2)-C and Kelly cells. Mass spectrometry analysis showed that lncNB1 RNA specifically bound to ribosomal protein L35 (RPL35), the RNA helicase DDX42, the histone protein H1X, interleukin enhancer-binding factor 2 (ILF2), and heterogeneous nuclear ribonucleoprotein K (HNRPK) proteins in both BE(2)-C and Kelly cells (Supplementary Table 4).

We then employed siRNAs to knock down RPL35, DDX42, H1X, ILF2, and HNRPK in Kelly cells, and found that only knocking down RPL35 significantly down-regulated DEPDC1B, N-Myc, and E2F1 protein expression (Fig. 5a, Supplementary Fig. 6a–c). RT-PCR analysis further showed that knocking down RPL35 with two independent siRNAs significantly reduced DEPDC1B, but not N-Myc and E2F1, mRNA expression (Supplementary Fig. 6c). Immunoblot analysis demonstrated that RPL35 siRNAs consistently reduced E2F1 and DEPDC1B protein

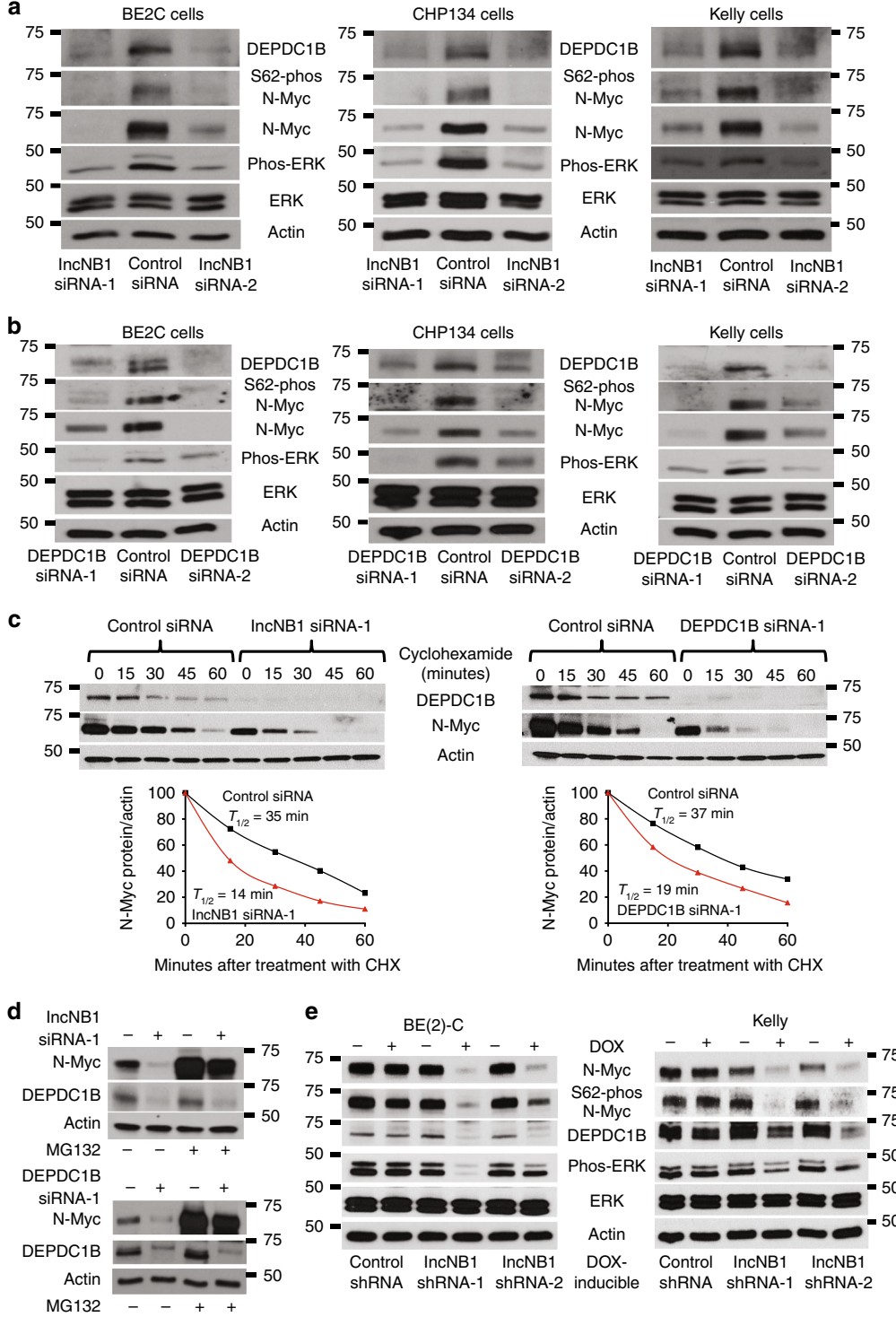

**Fig. 3** LncNB1 stabilizes N-Myc protein through DEPDC1B-mediated ERK protein phosphorylation. **a**, **b** BE(2)-C, Kelly, and CHP134 cells were transfected with control siRNA, lncNB1 siRNA-1, lncNB1 siRNA-2 (**a**), DEPDC1B siRNA-1 or DEPDC1B siRNA-2 (**b**) for 48 h, followed by immunoblot analysis of DEPDC1B, ERK, phosphorylated (phos-) ERK, S62-phos-N-Myc, and total N-Myc protein expression. **c** BE(2)-C cells were transfected with control siRNA, lncNB1 siRNA-1, or DEPDC1B siRNA-1 for 30 h, and treated with 50 μM cycloheximide (CHX) for the last 15, 30, 45, or 60 min. Protein was extracted for immunoblot analysis of DEPDC1B and N-Myc. N-Myc protein level was normalized by actin, and N-Myc protein half-life ($T_{1/2}$) was obtained from the line chart. **d** BE(2)-C cells were transfected with control siRNA, lncNB1 siRNA-1, or DEPDC1B siRNA-1 for 48 h, followed by treatment with MG-132 (10 μM) for 3 h. DEPDC1B and N-Myc protein expression was analyzed by immunoblot. **e** Doxycycline (DOX)-inducible control shRNA, lncNB1 shRNA-1 or lncNB1 shRNA-2 BE(2)-C and Kelly cells were treated with vehicle control or DOX for 48 h, followed by immunoblot analysis of DEPDC1B, phos-ERK, total ERK, S62-phos-N-Myc, or total N-Myc protein expression. Source data are provided as a Source Data file

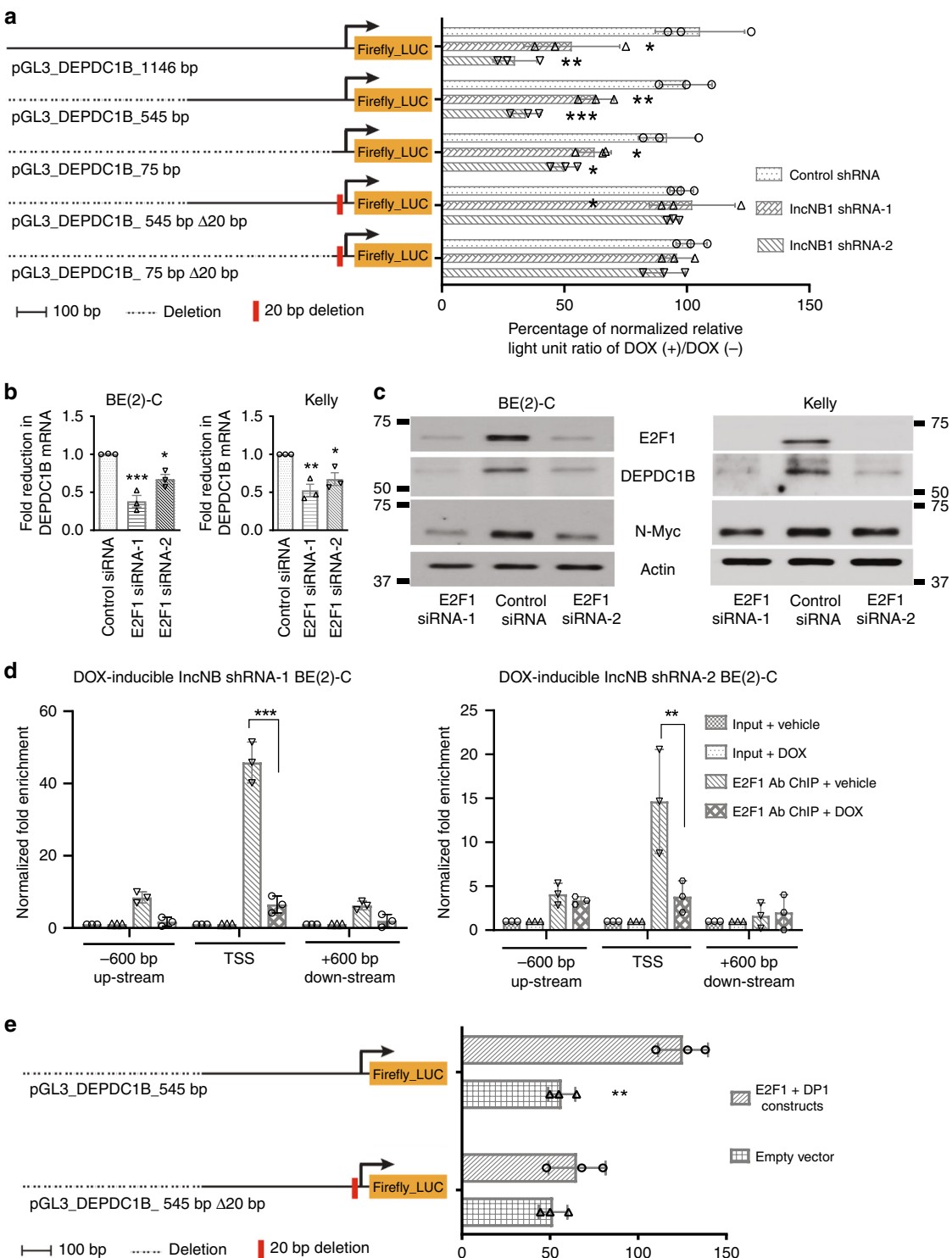

expression, ERK protein phosphorylation, N-Myc protein phosphorylation at S62 and expression (Fig. 5a). RNA immuno-precipitation assays showed that an anti-RPL35 antibody, compared with a control rabbit IgG, efficiently pulled down RPL35 protein in BE(2)-C and Kelly cells (Fig. 5b), that RPL35 bound to lncNB1 RNA and E2F1 RNA (Fig. 5c, d), and that transfection with lncNB1 siRNA-1 or siRNA-2 reduced RPL35 protein binding to E2F1 RNA (Fig. 5d).

BE(2)-C and Kelly cells were next transfected with control siRNA, lncNB1 siRNAs, or RPL35 siRNAs, followed by puromycin

incorporation assays for measuring protein synthesis[25]. Immuno-precipitation with an anti-puromycin antibody showed that the anti-puromycin antibody efficiently immunoprecipitated puromycin-labeled proteins, that lncNB1 siRNAs reduced the synthesis of E2F1 protein but not global proteins (Fig. 5e), and that RPL35 siRNAs considerably reduced the synthesis of proteins globally, including E2F1 protein (Fig. 5f). BE(2)-C cells were then transfected with control siRNA, lncNB1 siRNA-1, or lncNB1 siRNA-2, followed by cycloheximide treatment and polysome fractionation. Polysome profiling and RT-PCR showed

**Fig. 4** LncNB1 activates *DEPDC1B* gene transcription through the transcription factor E2F1. **a** Wild type and E2F1-binding site deletion mutant (Δ20bp) DEPDC1B gene promoters were cloned into a pGL3 firefly-luciferase (Firefly_LUC) construct. DOX-inducible control shRNA, lncNB1 shRNA-1, and lncNB1 shRNA-2 BE(2)-C cells were transfected with the wild type or Δ20bp deletion mutant DEPDC1B gene promoter pGL3 constructs, followed by treatment with vehicle control or DOX and luciferase assays. Percentage change in luciferase activity, measured as relative light unit (RLU) due to DOX treatment as compared with vehicle control treatment, was normalized by the luciferase activity of DOX-inducible control shRNA cells treated with vehicle control. Data were shown as the mean ± standard deviation, and evaluated by one-way ANOVA. *, **, and *** indicated $P < 0.05$, 0.01, and 0.001 respectively. **b**, **c** BE(2)-C and Kelly cells were transfected with control siRNA, E2F1 siRNA-1, or E2F1 siRNA-2 for 48 h, followed by RT-PCR analysis of DEPDC1B (**b**) and immunoblot analysis of DEPDC1B, E2F1, and N-Myc (**c**). Data were shown as the mean ± standard error of three independent experiments, and evaluated by one-way ANOVA. *, **, and *** indicated $P < 0.05$, 0.01, and 0.001 respectively. **d** ChIP assays were performed with a specific anti-E2F1 antibody (Ab) in DOX-inducible lncNB1 shRNA-1 or shRNA-2 BE(2)-C cells after treatment with vehicle control or DOX, followed by PCR with primers targeting different regions of the *DEPDC1B* gene promoter (−600 bp, transcription start site (TSS) or +600 bp). **e** BE(2)-C cells were co-transfected with pCMV14-empty vector or pCMV14-E2F1 plus pCMV10-3 × Flag-DP1 constructs in combination with wild type or E2F1-binding site (Δ20bp) deletion mutant *DEPDC1B* gene promoter pGL3 Firefly_LUC construct. In **d**, **e** data were shown as the mean ± standard deviation of three independent experiments, and evaluated by two-sided unpaired Student's *t*-test. ** and *** indicated $P < 0.01$ and 0.001, respectively. Source data are provided as a Source Data file

that lncNB1 knockdown shifted E2F1 mRNA (Fig. 5g), but not N-Myc mRNA (Supplementary Fig. 6d), from heavy polysomes to light polysomes, indicating reduced E2F1 mRNA translation. In addition, Affymetrix microarray experiments revealed that lncNB1 knockdown resulted in reduction in the expression of 34 mRNAs, including E2F1 mRNA, in the heavy polysomes (Supplementary Table 5), and fluorescence in situ hybridization and immunocytochemistry double labeling experiments showed that lncNB1 RNA and RPL35 protein were mostly located in the cytoplasm in neuroblastoma cells (Supplementary Fig. 7a, b).

Taken together, the data demonstrate that lncNB1 increases E2F1 protein expression by binding to the ribosomal protein RPL35, leading to E2F1 protein synthesis and DEPDC1B gene transcription.

**LncNB1 induces neuroblastoma cell proliferation and survival**. We next investigated if lncNB1, DEPDC1B, RPL35, or E2F1 is required for neuroblastoma cell proliferation and/or survival. BE (2)-C, Kelly, and CHP134 cells were transfected with control siRNA, lncNB1 siRNA-1, or lncNB1 siRNA-2. Alamar blue assays showed that knocking down lncNB1 expression reduced the numbers of viable cells (Fig. 6a), BrdU incorporation assays showed that lncNB1 knockdown consistently reduced neuroblastoma cell proliferation (Supplementary Fig. 8), and flow cytometry analysis of Annexin V positively stained cells showed that knocking down lncNB1 induced apoptosis (Fig. 6b), suggesting that lncNB1 is required for neuroblastoma cell proliferation and survival. Consistent with the data from lncNB1 siRNAs, transfection with DEPDC1B siRNAs, RPL35 siRNAs, or E2F1 siRNAs (Fig. 6c–e) also reduced the number of viable BE(2)-C and Kelly cells.

Doxycycline-inducible control shRNA, lncNB1 shRNA-1 or shRNA-2 BE(2)-C and Kelly cells were then treated with vehicle control or doxycycline, followed by analysis of the cell cycle and apoptosis. Cell cycle analysis showed that knocking down lncNB1 with doxycycline consistently reduced the percentage of neuroblastoma cells at the S phase (Fig. 6f), and apoptosis analysis showed that knocking down lncNB1 resulted in apoptosis (Fig. 6g). Taken together, the data suggest that lncNB1, its binding partner RPL35 and their targets E2F1 and DEPDC1B are required for neuroblastoma cell proliferation and survival.

**LncNB1 knockdown leads to neuroblastoma regression in mice**. We next examined whether lncNB1 is essential for neuroblastoma cell clonogenic capacity in vitro and tumor progression in vivo. Doxycycline-inducible control shRNA, lncNB1 shRNA-1 or lncNB1 shRNA-2 BE(2)-C and Kelly cells were treated with vehicle control or doxycycline. Clonogenic assays showed that treatment with doxycycline did not have an effect on colony formation in doxycycline-inducible control shRNA cells, but

almost completely abolished and indeed completely abolished clonogenic capacity of doxycycline-inducible lncNB1 shRNA-1 and lncNB1 shRNA-2 BE(2)-C and Kelly cells, respectively (Fig. 7a, b), suggesting that lncNB1 plays a critical role in neuroblastoma tumorigenesis.

Doxycycline-inducible lncNB1 shRNA-1 BE(2)-C cells were then xenografted into nude mice. When tumors reached 50 mm³, the mice were divided into two groups, and fed with food with or without doxycycline. As shown in Fig. 7c, d, treatment with doxycycline, compared with vehicle control, significantly suppressed neuroblastoma progression. The tumors shrunk over the first 10 days in all mice, then stabilized at approximately 220 mm³ in one mouse and completely regressed in four out of ten mice, over a period of 6 months. Survival curve showed that knocking down lncNB1 considerably improved mouse overall survival. These results therefore demonstrate that lncNB1 plays an essential role in *MYCN*-amplified neuroblastoma tumorigenesis.

**High lncNB1 in tumor tissues predicts poor patient prognosis**. To assess the clinical relevance of the above findings, we examined lncNB1, DEPDC1B, RPL35, and E2F1 expression in 493 human neuroblastoma tissues in the publicly available neuroblastoma tissue RNA sequencing-patient prognosis SEQC-RPM-seqcnb1 dataset downloaded from the *R2 platform* [http://r2.amc.nl] (last accessed on 26 August 2016)[26]. Five out of 498 samples were excluded from analysis, due to the lack of information on *MYCN* amplification status. Two-sided Pearson's correlation study and Student's *t*-test showed that lncNB1 RNA expression positively correlated with *MYCN* amplification (Fig. 8a), N-Myc mRNA expression (Supplementary Fig. 9a) as well as DEPDC1B mRNA expression (Fig. 8b), and that RPL35 and E2F1 mRNA expression also positively correlated with DEPDC1B mRNA expression in the 493 human neuroblastoma tissues (Fig. 8c, d).

Using the median level of RNA expression as the cut-off point, Kaplan–Meier survival analysis showed that high levels of lncNB1, DEPDC1B, RPL35, and E2F1 RNA expression in the 493 neuroblastoma tissues of the SEQC-RPM-seqcnb1 dataset were associated with poor patient prognosis (Fig. 8e–h). In addition, high levels of lncNB1, DEPDC1B, RPL35, or E2F1 RNA expression in 181 stage 4 neuroblastoma tissues of the SEQC-RPM-seqcnb1 dataset were also associated with poor patient prognosis (Fig. 8i–l). Consistent with these data, high levels of lncNB1, DEPDC1B, RPL35, or E2F1 RNA expression in 88 neuroblastoma tissues of the much smaller publicly available microarray gene expression-patient prognosis Versteeg dataset, which was also downloaded from the R2 platform, were also associated with poor patient prognosis (Supplementary Fig. 9b–e).

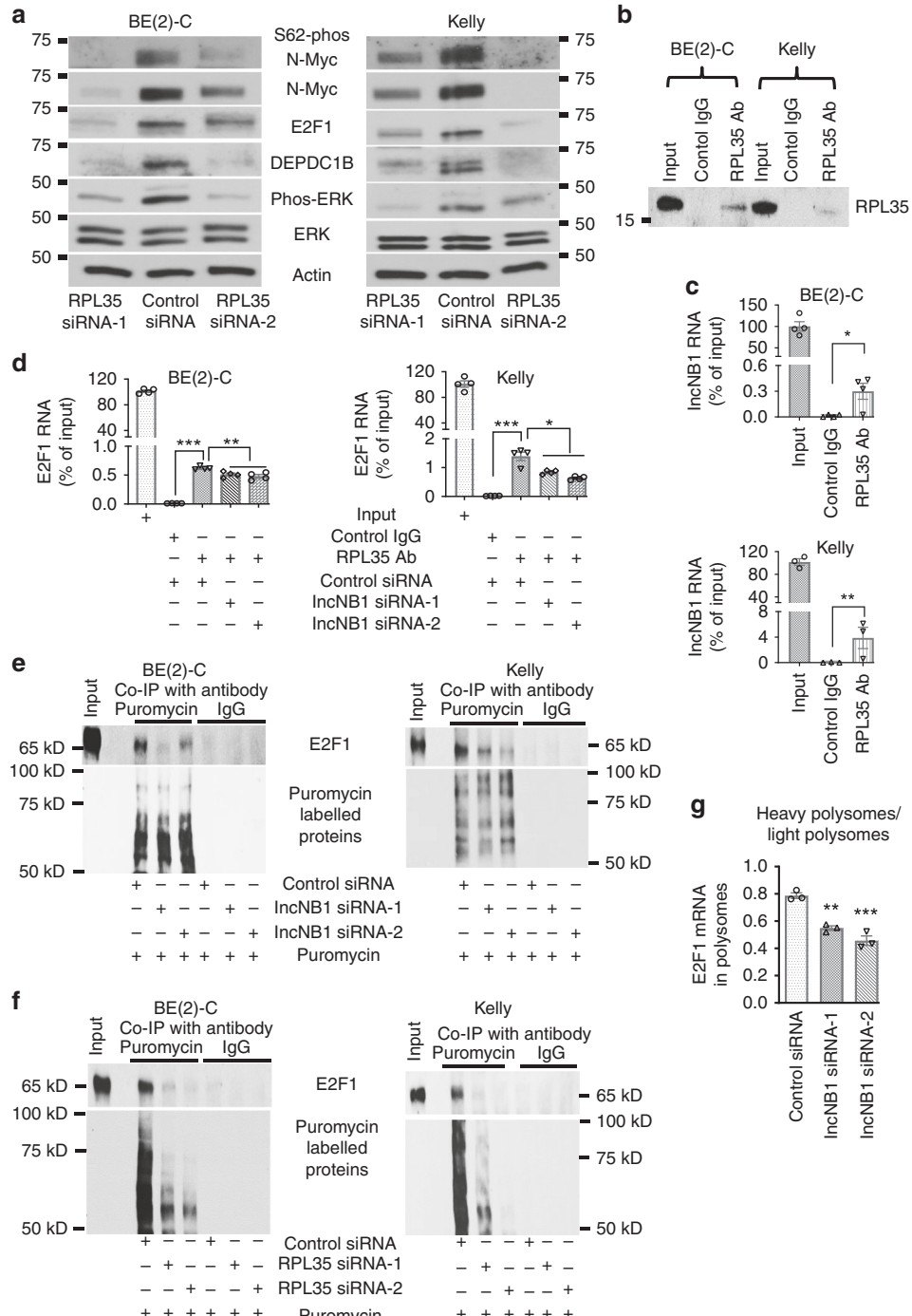

**Fig. 5** LncNB1 increases E2F1 protein translation by binding to the ribosomal protein RPL35. **a** BE(2)-C and Kelly cells were transfected with control siRNA, RPL35 siRNA-1, or RPL35 siRNA-2 for 48 h, followed by immunoblot analysis of DEPDC1B, ERK, phosphorylated ERK (phos-ERK), N-Myc, S62-phos-N-Myc, and E2F1 protein expression. **b**–**d** RNA immunoprecipitation assays were performed with a control IgG or an anti-RPL35 antibody (Ab) without siRNA transfection (**b**, **c**) or after transfection with control siRNA, lncNB1 siRNA-1, or lncNB1 siRNA-2 (**d**) in BE(2)-C and Kelly cells. Immunoprecipitated protein was immunoblotted with the anti-RPL35 antibody (**b**), and immunoprecipitated RNA was examined by RT-PCR with primers targeting lncNB1 (**c**) or E2F1 (**d**). **e**, **f** BE(2)-C and Kelly cells were transfected with control siRNA, lncNB1 siRNA-1 or siRNA-2 (**e**) or RPL35 siRNA-1 or RPL35 siRNA-2 (**f**) for 48 h, followed by labeling with puromycin (10 μg/ml) for 15 min. Co-immunoprecipitation (Co-IP) was performed with a control IgG or anti-puromycin antibody and immunoblot with anti-E2F1 and anti-puromycin antibodies. **g** BE(2)-C cells were transfected with control siRNA, lncNB1 siRNA-1, or lncNB1 siRNA-2 for 48 h, followed by treatment with 50 μg/ml of cycloheximide and polysome fractionation. RT-PCR analysis of E2F1 mRNA was performed and results were pooled into light (fraction 7, 8, and 9) and heavy (fraction 12, 13, and 14) polysomes and expressed as the ratio of mRNA in heavy polysomes and mRNA in light polysomes. In **c**, **d**, **g** data were shown as the mean ± standard error of three or four independent experiments, and evaluated by two-sided unpaired Student's $t$-test or one-way ANOVA. *, **, and *** indicate $P < 0.05$, 0.01, and 0.001, respectively. Source data are provided as a Source Data file

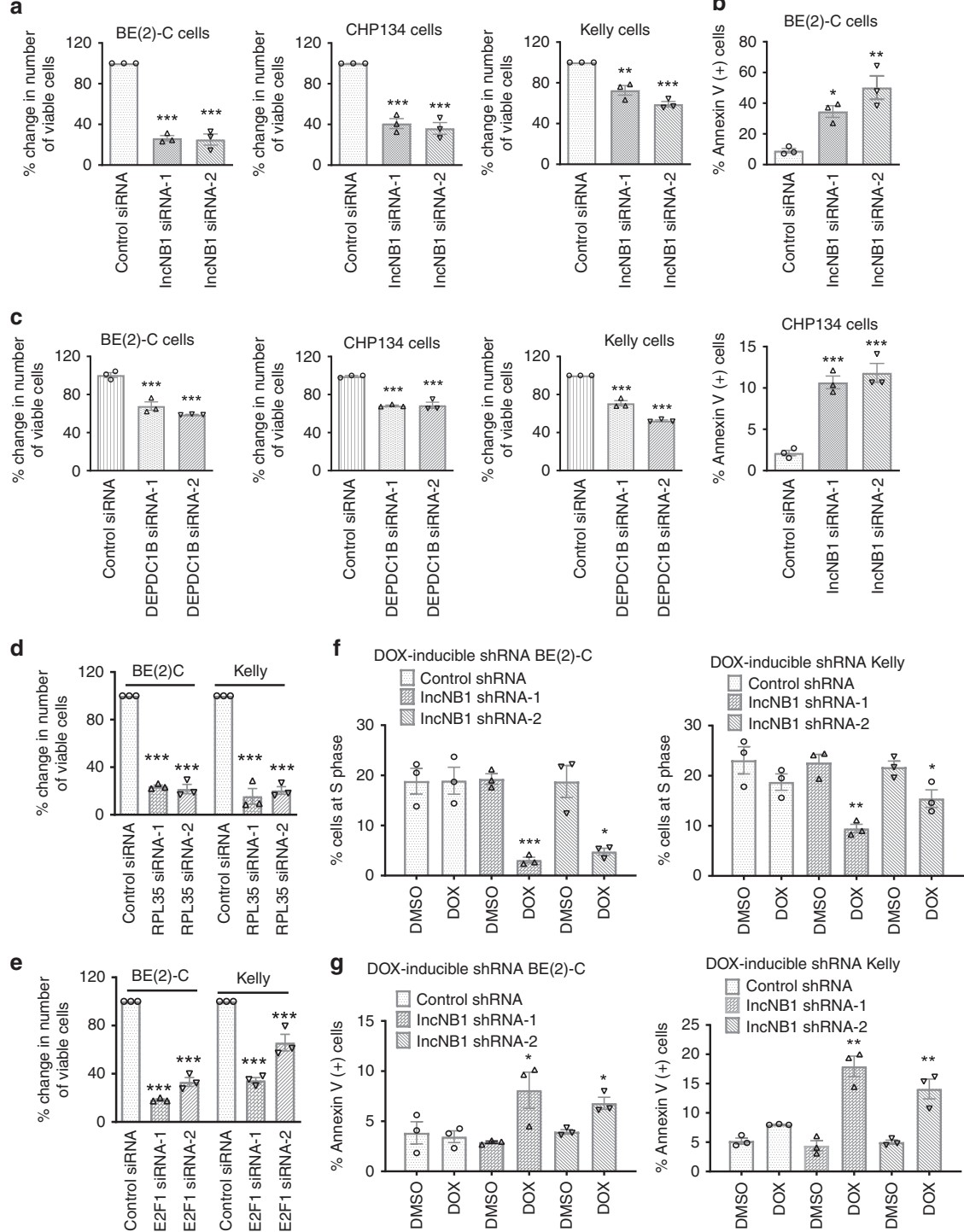

**Fig. 6** LncNB1 is required for neuroblastoma cell proliferation and survival. **a** BE(2)-C, Kelly, and CHP134 cells were transfected with control siRNA, lncNB1 siRNA-1, lncNB1 siRNA-2 for 96 h. Relative numbers of viable cells were examined by Alamar blue assays and expressed as percentage changes. **b** BE(2)-C and CHP134 cells were transfected with control siRNA, lncNB1 siRNA-1, or lncNB1 siRNA-2 for 96 h, followed by staining with Annexin V and flow cytometry analysis of Annexin V positively stained cells. **c–e** BE(2)-C, CHIP134, and/or Kelly cells were transfected with control siRNA, DEPDC1B siRNA-1, or siRNA-2 (**c**), RPL35 siRNA-1 or siRNA-2 (**d**), or E2F1 siRNA-1 or siRNA-2 (**e**) for 96 h. Relative numbers of viable cells were examined by Alamar blue assays and expressed as percentage changes. **f** DOX-inducible control shRNA, lncNB1 shRNA-1 or lncNB1 shRNA-2 BE(2)-C and Kelly cells were treated with vehicle control or DOX for 96 h, followed by staining with propidium iodide and flow cytometry analysis of the cell cycle. The percentage of cells at the S phase was shown. **g** DOX-inducible control shRNA, lncNB1 shRNA-1 or lncNB1 shRNA-2 BE(2)-C and Kelly cells were treated with vehicle control or DOX for 96 h, followed by staining with Annexin V and flow cytometry analysis of Annexin V-positive cells. Throughout, data were shown as the mean ± standard error of three independent experiments, and evaluated by two-sided unpaired Student's $t$-test for two groups or one-way ANOVA for more than two groups. *, **, and *** indicate $P < 0.05$, 0.01, and 0.001, respectively. Source data are provided as a Source Data file

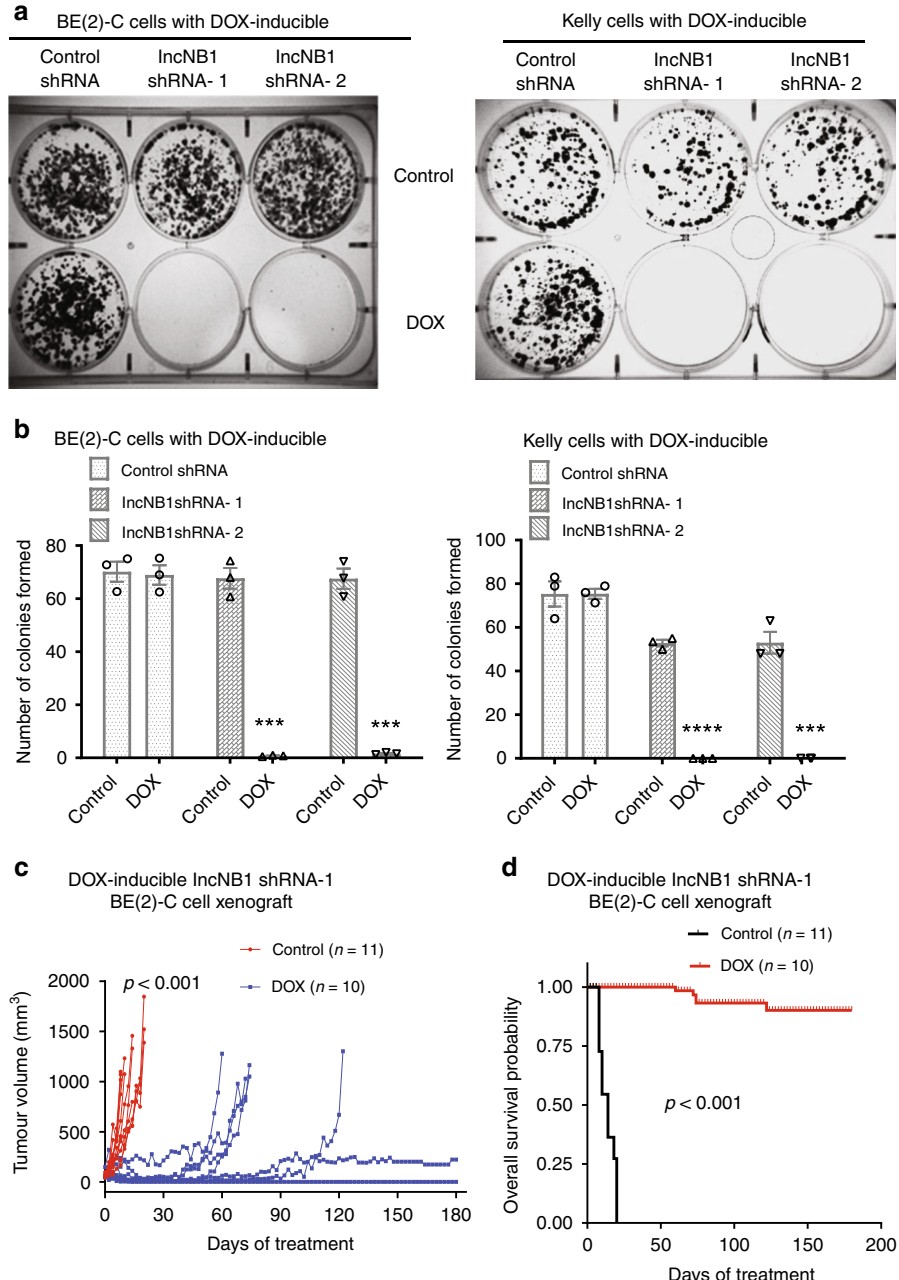

**Fig. 7** LncNB1 knockdown leads to tumor regression in neuroblastoma-bearing mice. **a**, **b** DOX-inducible control shRNA, lncNB1 shRNA-1 or shRNA-2 BE (2)-C and Kelly cells were treated with vehicle control or DOX for 14 [BE(2)-C cells] or 18 (Kelly cells) days. Cell colonies were fixed with crystal violet (**a**) and the numbers of colonies counted (**b**). Data were shown as the mean ± standard error of three independent experiments, and evaluated by two-tailed unpaired Student's $t$-test. ***$P < 0.001$. **c**, **d** DOX-inducible lncNB1 shRNA-1 BE(2)-C cells were xenografted into nude mice. When the tumor reached 0.05 cm³, the mice were divided into two groups and given feed with or without DOX. Tumor growth was monitored, and the mice were culled when tumor size reached 1.0 cm³ or 6 months later. The effect of DOX on tumor progression was evaluated with two-way ANOVA (**c**). The survival curve showed the probability of overall survival of the mice, and $P$ value was obtained from two-sided log-rank test (**d**). Source **d**ata are provided as a Source Data file

Taken together, the data suggest that lncNB1, RPL35, and E2F1 regulate DEPDC1B expression in human neuroblastoma tissues, and that high levels of lncNB1, DEPDC1B, RPL35, and E2F1 expression in tumor tissues predict poor prognosis in neuro-blastoma patients.

## Discussion

Protein-coding genes important for c-Myc and N-Myc-induced tumorigenesis have been the focus of intense research in the past three decades. In comparison, little is known about lncRNAs in Myc-induced carcinogenesis. Here our RNA sequencing analysis identifies N-Myc, MYCNOS, IGF2BP1, lncNB1, RNF217, RP11-102F4.3, GRIK3, and SLCO5A1 as the RNAs most considerably over-expressed in *MYCN*-amplified, compared with *MYCN*-non-amplified, neuroblastoma cell lines. MYCNOS[18] and IGF2BP1[16] promote *MYCN*-driven neuro-blastoma metastasis and tumorigenesis, respectively, RNF217[27], GRIK3[28], and SLCO5A1[29] promotes leukemia, breast and colon

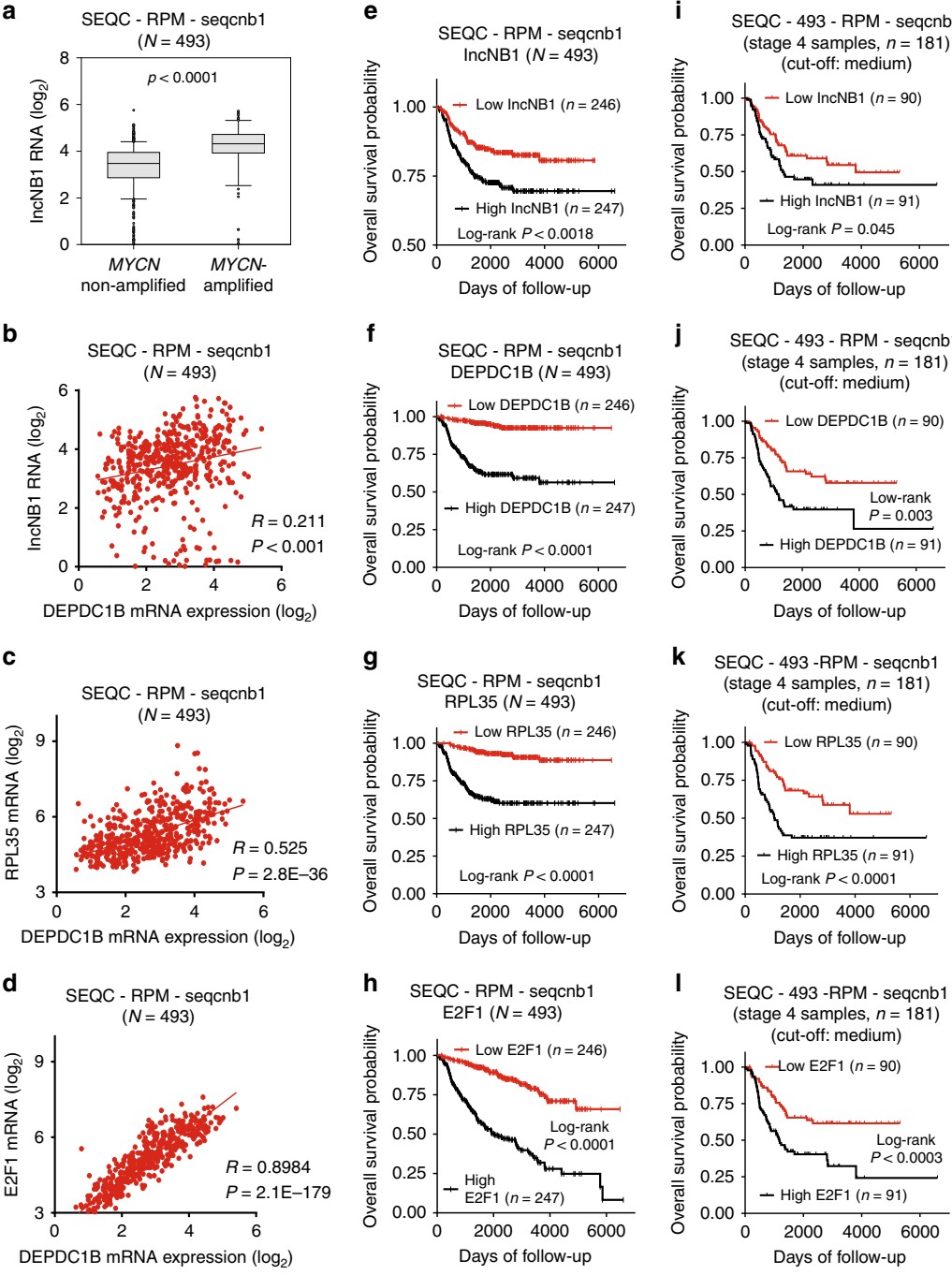

**Fig. 8** High lncNB1, DEPDC1B, E2F1, and RPL35 in tumor tissues predict poor patient prognosis. **a**, **b** Two-sided unpaired Student's *t*-test and two-sided Pearson's correlation were employed to analyze the association between lncNB1 RNA expression and *MYCN* gene amplification (**a**) or DEPDC1B mRNA expression (**b**), respectively in 493 human neuroblastoma tissues in the publicly available RNA sequencing gene expression-patient prognosis SEQC-RPM-seqcnb1 dataset, downloaded from the *R2 platform* [http://r2.amc.nl]. In the box and whisker plot (**a**), the center line was the median, the ends of the box were the upper and lower quartiles, and the whiskers extended to the highest and the lowest 10% values. **c**, **d** Two-sided Pearson's correlation was employed to analyze the correlation between RPL35 and DEPDC1B mRNA expression (**c**), as well as between E2F1 and DEPDC1B mRNA expression (**d**), in the SEQC-RPM-seqcnb1 dataset. **e**–**h** Kaplan–Meier curves showed the probability of overall survival of neuroblastoma patients according to the levels of lncNB1 (**e**), DEPDC1B (**f**), RPL35 (**g**), and E2F1 (**h**) expression in tumor tissues in the SEQC-RPM-seqcnb1 dataset, using the median of RNA expression as the cut-off points and two-sided log-rank test. **i**–**l** Kaplan–Meier curves showed the probability of overall survival of 181 stage 4 neuroblastoma patients according to the levels of lncNB1 (**i**), DEPDC1B (**j**), RPL35 (**k**), and E2F1 (**l**) in the SEQC-RPM-seqcnb1 dataset, using the median RNA expression as the cut-off points and two-sided log-rank test. Source data are provided as a Source Data file

cancer. However, the lncRNAs lncNB1 and RP11-102F4.3 have never been studied in normal physiology or disease. Analysis of TCGA datasets with more than 10,000 human tumor tissues of various organ origins shows that lncNB1 is most abundantly

over-expressed in human neuroblastoma tissues, suggesting a role of lncNB1 in neuroblastoma tumorigenesis.

LncRNAs are emerging as critical regulators of gene expression through transcriptional, post-transcriptional and post-translational

mechanisms. For example, several sense lncRNAs up-regulate the transcription of their neighboring protein-coding genes through interacting with the histone H3K4 methylation adaptor WDR5 or the transcriptional co-activator mediator[10,22,30], whereas the lncRNA NORAD sequesters PUMILIO proteins for genomic stability[14]. In this study, our genome-wide differential gene expression and gene set enrichment analysis show that knocking down lncNB1 considerably reduces E2F1 signaling activity and the expression of DEPDC1B, which is known to induce ERK protein phosphorylation[24]. While N-Myc protein is well-known to be stabilized after phosphorylation at S62 by phosphorylated ERK protein[3,4], we have found that knocking down lncNB1 or DEPDC1B consistently reduces DEPDC1B mRNA and protein expression, ERK protein phosphorylation, N-Myc protein phosphorylation at S62, expression and half-life, while N-Myc mRNA expression is not altered. Additionally, forced DEPDC1B over-expression reverses ERK protein dephosphorylation, N-Myc protein dephosphorylation at S62, and N-Myc protein reduction due to lncNB1 knockdown. Our data therefore suggest that lncNB1 up-regulates N-Myc protein expression through inducing DEPDC1B gene expression, leading to ERK protein phosphorylation and N-Myc protein stabilization, and that lncNB1 and DEPDC1B are factors important for N-Myc protein stability.

In this study, our RNA fractionation experiments show that lncNB1 RNA is located predominantly in the cytoplasm, suggesting that lncNB1 indirectly induces DEPDC1B gene transcription. Luciferase assays show that a 20 bp region enriched of E2F1-binding sites at the DEPDC1B gene core promoter is essential for lncNB1-induced DEPDC1B gene promoter activity, that E2F1 directly induces DEPDC1B gene promoter activity, and that the effect of E2F1 is abolished when the 20 bp region enriched of E2F1-binding sites is deleted. In concordance, ChIP assays confirm that E2F1 protein binding to the DEPDC1B gene core promoter is reduced with lncNB1 knockdown. Moreover, E2F1 siRNAs reduce DEPDC1B mRNA and protein expression and replicate the effects of lncNB1 and DEPDC1B siRNAs on ERK phosphorylation, N-Myc protein phosphorylation and expression. These data therefore suggest that lncNB1 indirectly induces DEPDC1B gene transcription through increasing protein expression of E2F1, which directly binds to the DEPDC1B gene promoter and enhances DEPDC1B gene transcription, leading to ERK protein phosphorylation and N-Myc protein stabilization.

Our RNA-binding protein pull-down and mass spectrometry analyses have identified the ribosomal protein RPL35, a component of the 60S ribosomal subunit, as one of the few proteins bound to lncNB1 RNA. In the literature, 60S ribosomal subunit proteins play key roles in mRNA translation and protein synthesis. For example, as a regulatory component of the ribosome, RPL13 binds to Rig-I and controls NF-κb protein synthesis[31], and RPL38 facilitates 80S complex formation on a select subset of Homeobox mRNAs to confer transcript-specific translational control[32,33]. RPL35 has recently been found to interact with eukaryotic translation elongation factor 2 and thereby control CSN2 protein synthesis[34]. In this study, our puromycin incorporation and polysome profiling assays demonstrate that knocking down lncNB1 considerably reduces E2F1 protein synthesis, and that knocking down RPL35 globally reduces protein synthesis. In addition, RPL35 siRNAs replicate the effects of lncNB1 siRNAs, DEPDC1B siRNAs, and E2F1 siRNAs on DEPDC1B expression, ERK protein dephosphorylation, N-Myc protein dephosphorylation at S62, and N-Myc protein reduction. Our data therefore suggest that lncNB1 RNA binds to RPL35 protein to facilitate E2F1 protein synthesis, leading to DEPDC1B gene transcription, ERK protein phosphorylation and consequent N-Myc protein phosphorylation at S62 and stabilization.

The role of the lncRNA lncNB1 in cancer or normal physiology is completely unknown. RPL35 has recently been found to be over-expressed in human colorectal cancer tissues with unknown functions[35]. We have found that knocking down lncNB1 expression results in neuroblastoma cell growth inhibition and apoptosis, abolishes neuroblastoma cell clonogenic capacity in vitro, and leads to tumor regression in neuroblastoma-bearing mice. Consistent with these data, DEPDC1B, E2F1, and RPL35 are also required for neuroblastoma cell proliferation and/or survival. In addition, in human neuroblastoma tissues, lncNB1, E2F1, or RPL35 RNA expression positively correlates with DEPDC1B RNA expression, and high levels of lncNB1, E2F1, RPL35, or DEPDC1B expression predict poorer patient outcome. Thus, our data demonstrate that lncNB1, its binding protein RPL35 and their target protein E2F1 and target gene DEPDC1B induce neuroblastoma cell proliferation and survival. As high levels of lncNB1, RPL35, E2F1, and DEPDC1B expression in tumor tissues correlate with poor prognosis in neuroblastoma patients, our findings identify lncNB1, RPL35, and DEPDC1B as important co-factors for N-Myc-driven oncogenesis and provide therapeutic targets for neuroblastoma.

## Methods

**Cell culture.** Neuroblastoma BE(2)-C, IMR32, SY5Y, and SHEP cells were cultured in Dulbecco's modified Eagle's medium (DMEM) supplemented with 10% fetal calf serum. SK-N-DZ and SK-N-AS cells were cultured in DMEM supplemented with 10% fetal calf serum and 1% non-essential amino acids. Kelly and CHP134 cells were cultured in Roswell Park Memorial Institute Medium (RPMI) 1640 supplemented with 10% fetal calf serum and 1% L-glutamine. BE(2)-C, SHEP, and SY5Y cells were provided by Barbara Spengler (Fordham University, New York, NY) 20 years ago. IMR32 were obtained from the American Type Culture Collection (Manassas, VA) 20 years ago. The Lenti-X™ 293T viral packaging cell line was purchased from Scientifix (South Yarra, Victoria, Australia). Kelly, CHP134, SK-N-DZ, and SK-N-AS cells were obtained from the European Collection of Cell Cultures through Sigma in 2010 (Sigma, Sydney, Australia). All cell lines were maintained in a humidified incubator at 37 °C and 5% $CO_2$ in air. The identity of cell lines was verified in 2010, 2014, 2015, 2016, 2017, and 2018 by short tandem repeat profiling conducted at the Garvan Institute of Medical Research or Cellbank Australia.

**Doxycycline-inducible shRNA expression cell lines.** The lentiviral doxycycline-inducible GFP-IRES-shRNA FH1tUTG construct from Dr. Marco Herold[36] was used to generate control shRNA and lncNB1 shRNA expressing constructs as well as neuroblastoma cell lines stably expressing the constructs. lncNB1 shRNA target sequences were GCTGCAGCGTTTACCCAAAGA (shRNA-1) and GCTTCCTTCAAACCTCAAATC (shRNA-2) (Supplementary Table 6). Sense and antisense shRNA oligoes were synthesized by GeneWorks (Thebarton, SA, Australia), and cloned into the doxycycline-inducible GFP-IRES-shRNA FH1tUTG construct. The doxycycline-inducible GFP-IRES-control shRNA, lncNB1 shRNA-1, or lncNB1 shRNA-2 FH1tUTG construct was transfected into 293T cells. Viral media were collected and employed to infect neuroblastoma cells with polybrene (Santa Cruz Biotechnology, Santa Cruz, CA) for 72 h. Fluorescence-activated cell sorting was performed with BD FACS Jazz™ II Cell Sorter (BD Biosciences, Franklin Lakes, NJ) to select neuroblastoma cells with high GFP protein expression. Cells were treated with 2 µg/ml doxycycline (Sigma) or DMSO vehicle control every 24 h to or to not induce shRNA expression.

**siRNA transfection.** Target sequences of siRNAs were 5′-CAGCTGCAGCGTTTACCCAAA-3′ (siRNA-1) and 5′-CACAGCGAATGCTAACTGATA-3′ (siRNA-2) (Qiagen, Hamburg, Germany) for lncNB1; 5′-GAGGAGCGTGTGGCTCATCTA-3′ (siRNA-1) (Qiagen) and 5′-GAGTTATTAGCTGCTAGATTGGTAA-3′ (siRNA-2) (Invitrogen, Carlsbad, CA) for DEPDC1B; 5′-CGTGCCGGAGTTGGTAAAGAA-3′ (siRNA-1) and 5′-TCCAGCGAGCTGATCCTCAAA-3′ (siRNA-2) (Qiagen) for N-Myc; 5′-TGGCATTTGTATTGATAGTTA-3′ (siRNA-1) and 5′-TATGTTAGTTGTGAAGAACTA-3′ (siRNA-2)(Qiagen) for HNRPK; 5′-CTCCATAGAAGTGTCATTCCA-3′ (siRNA-1) and 5′-CAGGCCCTTTGTACCACATAT-3′ (siRNA-2) (Qiagen) for ILF2; 5′-CTCGTCGCTGGCCAAGATCTA-3′ (siRNA-1) and 5′-CGCCGAGGGAATGGCCAAGAA-3′ (siRNA-2) (Qiagen) for H1X; 5′-CCCGGCGTCTGGTAGAATTTA-3′ (siRNA-1) and 5′-TACTAAGGTGTCACCTTATTA-3′ (siRNA-2) for DDX42; 5′-CAGGACCTTCGTAGCATTGCA-3′ (siRNA-1) and 5′-CTCACTGAATCTGACCACCAA-3′ (siRNA-2) for E2F1; 5′-CCGTGTTCTCACAGTTATTAA-3′ (siRNA-1) and 5′-TGCAGCAATGGCCAAGATCAA-3′ (siRNA-2) for RPL35 (Supplementary Table 6). Negative

control siRNAs did not target any human genes (All Stars Negative Control siRNA, Qiagen).

Neuroblastoma cells were transfected with siRNAs using Lipofectamine 2000 (Life Technologies, Grand Island, NY) according to the manufacturer's instructions and plated onto either 24- or 6-well plates or T25 flasks. Transfected cells were harvested for either RNA or protein for analysis.

**Plasmid transfection.** The pCMV6-Entry lncNB1 expression construct was custom synthesized by Origene (Origene, Rockville, MD). DP1 cDNA was amplified from BE(2)-C cells with Herculase II TAQ polymerase (Agilent Technologies, Santa Clara, CA) using the following primers: TTTAAGCTTgcaaaagatgccggtcta attgaag (forward); TTTTCTAGAgtgtcctcgtcattctcgttg (reverse). E2F1 coding sequence was amplified from the HA-E2F1 pRcCMV construct (Addgene plasmid #21667)[37], using the following primers: TTTAAGCTTatggccttggccgggggcccctg (pCMV14-HindIII E2F1 forward) and TTTGAATTCtcagaaatccagggggtgaggt ccccaaag (pCMV14-ECORI E2F1 reverse). PCR amplicons were then cloned into pCMV10 or pCMV14 (Sigma, St Louis, MO), respectively, to generate pCMV10_3× Flag-DP1 and pCMV14_E2F1 constructs. Cells were transiently transfected with constructs using Lipofectamine 2000 reagent according to the manufacturer's protocol.

**Real-time reverse transcription PCR (RT-PCR).** RNA was extracted from cells with RNeasy Plus minikit (Qiagen) and quantified with a Nanodrop spectro- photometer (Thermo Fisher Scientific, Waltham, MA), according to the manu- facturer's instructions. cDNAs were then synthesized with Moloney murine leukemia virus reverse transcriptase (Invitrogen) and random hexonucleotide primers. RT-PCR was performed using gene- specific primers and Power SYBR Green Master Mix (Invitrogen) as the fluorescent dye in Applied Biosystems 7900 (Applied Biosystems, Grand Island, NY). No template controls were used to detect any non-specific amplification. The sequences of RT-PCR primers were 5′-CGACC ACAAGGCCCTCAGTA-3′ (forward) and 5′-CAGCCTTGGTGTTGGAGGAG-3′ (reverse) for N-Myc; 5′-AATACGCCAATGTCCTGCTC-3′ (forward) and 5′-TCA GTGCCTTGGCTTGTAGA-3′ (reverse) for lncNB1; 5′-AGCCTTGTTGGAGGAA GTCA-3′ (forward) and 5′-TTTCGGTTTGGGTTTTTCAG-3′ (reverse) for DEPDC1B; 5′-AATGAGGCTGCTGGTAACAAA-3′ (forward) and 5′-AAGGGTC CAGACCTGACAGA-3′ for TUBB2A; 5′-AGGACGGACAGACCCAGAC-3′ (forward) and 5′-CAATCCCATGCTCATCACTG-3′ (reverse) for TUBB2B; 5′-TC CGAATCCTCTCACATGGT-3′ (forward) and 5′-CCTTCTTCTCTGGTGGC TTC-3′ (reverse) for ILF2; 5′-GACGGCATGGTTGGTTTCA-3′ (forward) and 5′- ATTCTGATGGGCTCCATGTATCTAT-3′ (reverse) for HNRPK; 5′-TGGGC GCACCTACCTCAA-3′ (forward) and 5′-CCTGCAGAAGCGTGTCGTT-3′ for H1X; 5′-TGAGAACATGGATCGAGGAAATAA-3′ (forward) and 5′-TCTCCC ATAGCTCCTGTGGAA-3′ (reverse) for DDX42; 5′-GGACTCTTCGGAGAACTTTCAG AT-3′ (forward) and 5′-GGGCACAGGAAAACATCGAT-3′ (reverse) for E2F1; 5′-CGGTGCGGCCTCCAA-3′ (forward) and 5′-CACGGGCAATGGATTTCC-3′ (reverse) for RPL35; 5′-AGGCCAACCGCGAGAAG-3′ (forward) and 5′-ACAGC CTGGATAGCAACGTACA-3′ (reverse) for Actin. All primers were synthesized by Sigma (Sigma, Sydney, Australia). Following RT-PCR, the comparative threshold cycle (ΔΔCt) method[38] was used to evaluate fold changes in target genes, relative to the reference gene actin.

**Immunoblot.** Protein was extracted in RIPA buffer (150 mM NaCl, 1% NP-40, 0.5% sodium deoxycholate, 0.1% SDS, 50 mM Tris-Cl pH 7.5) containing protease inhibitors (Sigma, St Louis, MO) and phosphatase inhibitors (Roche, Penzberg, Germany). After centrifugation at $13,000 \times g$ at 4 °C for 20 min, supernatant was collected. Protein was quantified with the Bicinchoninic Acid Assay kit (Pierce, Rockford, IL). For immunoblot analysis, protein samples were loaded onto sodium dodecyl sulfate-polyacrylamide gels, followed by electrophoresis and transfer to nitrocellulose membranes. Membranes were blocked with 10% skim milk powder in phosphate-buffered saline (PBS), and probed with the following primary anti- bodies: rabbit anti-DEPDC1B (1:500) (HPA038255; Sigma), rabbit anti-S62- phospho N-Myc (1:500) (A300-206A; Bethyl Laboratories, Montgomery, TX), rabbit anti-T58-phospho c-Myc (1:2000) (ab28842; Abcam, Cambridge, MA), mouse anti-N-Myc (1:1000) (sc-53993; Santa Cruz Biotechnology), rabbit anti-p- p44/42 MAPK (T202/Y204) (phos-ERK) (1:500) (9101S; Cell Signalling, Danvers, MA) or (1:500) (ab50011; Abcam), rabbit anti-p44/42 MAPK (ERK) (1:500) (9102S; Cell Signalling), rabbit anti-RPL35 (1:500) (ab209087 or ab121244; Abcam), or rabbit anti-E2F1 (1:1000) (3742S; Cell Signalling). The membranes were then incubated with a goat anti-rabbit (sc-2004) or goat anti-mouse (sc-2005) antibody conjugated to horseradish peroxidase (1:10,000) (both from Santa Cruz Biotechnology), and protein bands were visualized with ECL (Biorad, Hercules, CA). The membranes were finally probed with an anti-actin antibody (1:15000) (A5441; Sigma) as loading controls.

**RNA-binding protein pull-down assays.** pCMV6-entry lncNB1 expression con- struct was custom synthesized by OriGene, digested, and the linearized product was purified using Qiagen PCR purification Kit (Qiagen). Full-length lncNB1 was synthesized using primers containing the T7 promoter sequence in the forward primer (taatacgactcactatagggagaTTCCTGTCATGTGAAACATG) and SP6

promoter sequence in the reverse primer (atttaggtgacactatagaagggGGCCAA- CAACTGTTTAATG) with pCMV6-entry lncNB1 expression construct as tem- plates. The full-length lncNB1 cDNA containing the T7 and SP6 promoters was in vitro transcribed into sense (experimental) or antisense lncNB1 RNA (negative control) with the Biotin RNA Labeling Mix kit (Roche) and T7 or SP6 polymerase, respectively. The in vitro-transcribed sense and antisense lncNB1 RNAs were treated with DNase I and purified with the RNeasy kit (Qiagen). Approximately $2 \times 10^7$ BE(2)-C and Kelly cells were harvested by scraping and resuspended in 1.2 ml of lysis buffer [150 mM NaCl, 50 mM Tris-Cl pH 7.5, 0.5% Triton X-100, 1× protease inhibitor cocktail and 100 U/ml of SUPERaseIN (Thermo Fisher Scien- tific)]. The cell lysate was sonicated using Bioruptor (Diagenode, Liege, Belgium) for 10 min with 30 s on/off cycles and centrifuged for 10 min at 4 °C at maximum speed. The protein lysate was pre-cleared with 50 μl of activated BcMag Monomer Avidin Magnetic beads (Bioclone, San Diego, CA), according to the manufacturer's protocol. Ethylenediaminetetraacetic acid was added to 20 μg of biotin-labeled sense or antisense lncNB1 RNA to a final concentration of 5 mM. The mixture was heated to 90 °C in RNA structure buffer (10 mM Tris pH 7, 0.1 KCl, 10 mM MgCl₂) for 2 min, and cooled at room temperature for 20 min to allow proper secondary structure formation. The biotin-labeled sense or antisense lncNB1 RNA was then added to the pre-cleared protein lysate and incubated overnight at 4 °C with rotation. Proteins which bound to sense or antisense lncNB1 RNA were isolated with BcMag Monomer Avidin Magnetic beads according to the manufacturer's protocol. The eluted proteins were examined by mass spectrometry and analyzed using label-free quantification in Scaffold 4. Candidate proteins were defined as those that were detectable with a minimum of two peptides in at least two experimental repeats in both BE(2)-C and Kelly cells, with at least 2.5-fold enrichment compared to the control antisense lncNB1 RNA pull down.

**A immunoprecipitation assays.** RNA immunoprecipitation assays were per- formed using Magna RIP Kit from Merck Millipore (Burlington, MA) according to the manufacturer's instructions, with 5 μg of control IgG or anti-RPL35 antibody (Abcam ab209087) for immunoprecipitation and primers targeting lncNB1 (5′-AA TACGCCAATGTCCTGCTC-3′ as the forward primer and 5′-TTTCCAGTGT CCTTCGAACC-3′ as the reverse primer), the positive control U1 SNRNP70 or the negative control U1 RNAs for RT-PCR.

**Affymetrix microarray gene expression study.** BE(2)-C cells were transfected with control siRNA, lncNB1 siRNA-1 or lncNB1 siRNA-2. Forty hours later, RNA was extracted from the cells with RNeasy Mini kit (Qiagen), and four replicates of differential gene expression was examined using Affymetrix Arrays (Affymetrix, Santa Clara, CA)[39] under two platforms (three replicates from HuGene-2_0-st and one from Clariom_S_Human). Results from the microarray experiment from the two platforms were analyzed in R [http://www.r-project.org/] with the bio- conductor package [http://www.bioconductor.org/], and normalized with RMA method (from R package oligo, Version 1.42.0) independently. Then, the meta- analysis was implemented with the R package RankProd[40,41] to do the downstream differential expression analysis based on the combined normalized data. Gene set enrichment analysis (GSEA) was conducted with the R package fgsea based on the results generated from RankProd[42] [pathway analysis against molecular signature database C3: Motif gene sets (Broad Institute, Cambridge, MA)]. The genes were ranked with the log₂ fold change values, and the false discovery rate-adjusted P values were computed with the Benjamini–Hochberg method[42].

In addition, BE(2)-C cells were transfected with control siRNA, lncNB1 siRNA- 1 or lncNB1 siRNA-2 for 48 hours, followed by treatment with 50 μg/ml of cycloheximide and polysome fractionation. RNA was extracted from the heavy polysome and subjected to Clariom_S_Human Affymetrix Array experiments (Affymetrix, Santa Clara, CA)[39]. Results from the microarray experiment were analyzed in R [http://www.r-project.org/] with bioconductor package [http://www. bioconductor.org/], and normalized with the RMA method (from R package oligo, version 1.42.0). The downstream differential expression analysis was conducted with the Limma package (version 3.34.9).

**RNA extraction and sequencing.** RNA was extracted from six human neuro- blastoma cell lines [BE(2)-C, Kelly, CHP134, SK-N-DZ, SK-N-AS and SY-5Y] using Direct-zol™ RNA kit (Zymo Research, Irvine, CA), purified with RNeasy Plus Mini kit (Qiagen), and removed of ribosomal RNA with Ribo-Zero™ rRNA Removal Magnetic kit (Illumina, San Diego, CA). RNA yield and quality were measured with a BioSpec-nano spectrophotometer (Shimadzu Scientific Instru- ments, Kyoto, Japan), and integrity was confirmed with Bioanalyzer separation chips (Agilent Technologies) with A260/A280 ratios of >2.0 for all samples and a minimum RNA Integrity Number (RIN) of 6.

One hundred nanograms of ribosome-depleted RNA was used as input to the TruSeq mRNA Sample Prep v2 kit (Illumina), without the poly-A pulldown step. Sample preparation was performed according to the manufacturer's instructions but starting with a 94 °C incubation at 2 min to elute, fragment and prime the RNA samples. PCR cycles were reduced from 15 to 12 cycles to minimize the duplication rate. The libraries were multiplexed into one lane and sequenced on a HiSeq2000 (Illumina) using TruSeq SBS Kit v3 kit (Illumina) and 100 bp paired-end reads.

**RNA sequencing data analysis**. The unique RNA sequencing reads of six neuroblastoma cell lines [SK-N-AS, SY5Y, CHP134, SK-N-DZ, Kelly, and BE(2)-C] were each aligned to Hg19 using STAR aligner[43]. The aligned reads were counted towards Gencode V.18 (ref. [44]) to quantify gene expression for all genes using the union model. Genes were considered expressed if they had an expression of ≥1 counts per million in at least two samples. The expression of all samples were normalized using the Trimmed Mean of $M$ values model, and differential expression analysis was performed using EdgeR[45], comparing the *MYCN*-amplified [CHP134, SK-N-DZ, Kelly, and BE(2)-C] versus the *MYCN*-non-amplified (SK-N-AS, SY5Y) cell lines. Genes with an absolute log2 fold change of >1 and a false discovery rate of <0.05 were considered differentially expressed.

**GTEx RNA sequencing data analysis**. The publicly available Genotype-Tissue Expression (GTEx) Release V7 dataset (dbGaP Accession phs000424.v7.p2) provides expression data of all transcripts from RNA sequencing of 53 normal tissue sites across nearly 1000 people. lncNB1 transcript expression levels in the normal tissues were directly obtained through GTEx Portal website [https://gtexportal.org/home/gene/RP1-40E16.9%20].

**TCGA RNA sequencing data analysis**. RNA sequencing data from TCGA was complemented with RNA sequencing data from 493 primary neuroblastoma tumors[26] and processed using Kallisto [https://pachterlab.github.io/kallisto/], applying a gene index based on Ensembl version 75, to quantify expression of lncNB1. The RNA sequencing data from TCGA were obtained from various cancers, including adrenocortical carcinoma (ACC), urothelial bladder cancer, breast cancer, cervical cancer, cholangiocarcinoma, colon and rectal adenocarcinoma, diffuse large B-cell lymphoma, esophageal cancer, glioblastoma multiforme, head and neck squamous cell carcinoma, chromophobe renal cell carcinoma, clear cell kidney carcinoma, papillary kidney carcinoma, low-grade glioma, liver hepatocellular carcinoma, lung adenocarcinoma, lung squamous cell carcinoma, mesothelioma, neuroblastoma, ovarian serous cystadenocarcinoma, pancreatic ductal adenocarcinoma, pheochromocytoma & paraganglioma, prostate adenocarcinoma, colon and rectal adenocarcinoma, sarcoma, cutaneous melanoma, testicular germ cell cancer, papillary thyroid carcinoma, thymoma, uterine corpus endometrial carcinoma, uterine carcinosarcoma, uveal melanoma.

**ChIP assays**. For analyzing histone H3K4 trimethylation at the *DEPDC1B* and *E2F1* gene promoters and for analyzing E2F1 protein binding at the *DEPDC1B* gene promoter, ChIP assays were performed with 5 ug of a rabbit anti-H3K4me3 (ab8580; Abcam), rabbit anti-E2F1 (3742S; Cell Signalling), or control rabbit antibody, followed by real-time PCR with primers targeting a remote negative control region or the core promoter region of the *DEPDC1B* or *E2F1* gene[46]. The sequences of primers used were: 5′-AGAAGTCTGGGAAGGGTGCT-3′ (forward) and 5′-ATGCCAGCTTCTTGAGCATT-3′ (reverse) for the negative control region of the DEPDC1B gene; 5′-GTTTCGGTCGCTGGATAACA-3′ (forward) and 5′-CTAGGCAGGTGCGACTAAGG-3′ (reverse) for the *DEPDC1B* gene promoter; 5′-GCAGATGAGGCAAGCAAAGC-3′ (forward) and 5′-CCATCCAAAAGGCAGTCTAACAT-3′ (reverse) for the negative control region of the *E2F1* gene; 5′-AATGTCATGGGTGAGGCAAGTT-3′ (forward) and 5′-CAACCTGTAGCCCCCAACAG-3′ (reverse) for the *E2F1* gene promoter. Fold enrichment of the *DEPDC1B* and *E2F1* gene promoter regions was calculated by dividing PCR products from samples immunoprecipitated by the experimental antibodies by PCR products from samples immunoprecipitated by the control antibody.

**Luciferase assays**. Modulation of *DEPDC1B* gene promoter activity by lncNB1 was analyzed by luciferase assays. pGL3 construct carrying *DEPDC1B* gene promoter (1146 bp) was obtained by directional cloning using primers as listed in Supplementary Table 6. Deletion mutants of pGL3_*DEPDC1B* gene promoter were obtained by whole-around PCR technology using primers as listed in Supplementary Table 6. Luciferase reporter activity was measured using the Dual Luciferase Assay System (Promega). Relative percentage luciferase activity of the doxycycline treatment condition was normalized by the luciferase activity of the same reporter construct under vehicle control treatment condition.

**Puromycin incorporation assays**. Neuroblastoma BE(2)-C and Kelly cells were transfected with control siRNA, lncNB1 siRNA-1, lncNB1 siRNA-2, RPL35 siRNA-1, or RPL35 siRNA-2 for 48 h, followed by treatment with vehicle control or with puromycin (10 μg/ml) for 15 min. For Kelly cells only, the medium was removed and replaced with fresh growth medium to chase for a further 1 h. Protein was then extracted for immunoblot analysis with mouse anti-puromycin (1:8000) (MABE343; Merck Millipore, Burlington, MA) and actin (1:10,000) (A5441; Sigma) antibodies, or co-immunoprecipitation with 5 μg of mouse control IgG2a (401501), K isotype (Biolegend San Diego, CA) or 5 μg of mouse anti-puromycin antibody (MABE343; Merck Millipore) with 700 μg of protein lysate and immunoblot analysis with anti-E2F1 (1:500) (3742S; Cell Signalling) and rat anti-puromycin (1:6000) (MABE341; Merck Millipore) antibodies.

**Polysome profiling**. BE(2)-C cells were transfected with control siRNA, lncNB1 siRNA-1, or lncNB1 siRNA-2 for 48 h, followed by treatment with 50 μg/ml of cycloheximide for 5 min. Cells were harvested by trypsinization, and cell pellets were washed with phosphate-buffered saline and then hypotonic solution (5 mM Tris pH 7.5, 1.5 mM potassium chloride, and 2.5 mM magnesium chloride) and lysed in hypotonic lysis buffer (5 mM Tris pH 7.5, 1.5 mM potassium chloride, 2.5 mM magnesium chloride, 0.5% (v/v) triton X-100, 0.5% (w/v) sodium deoxycholate, 30 mM dithiothreitol, 50 μg/ml cycloheximide, 1× ethylenediaminetetraacetic acid free protease inhibitor and 60 U/ml RNasin). Two hundred microliters of cell lysates containing 40 million cells/sample were loaded into ultracentrifuge tubes containing 10–40% sucrose gradient prepared with a Bio-Comp gradient maker and centrifuged in an ultracentrifuge with a SW41 rotor at 36,000 revolutions per minute. Polysomes were fractionated using Teledyne ISCO Foxy 21 Fractionation System (Teledyne ISCO, Lincoln, NE). Fractions were collected in equal volumes of Qiagen RNA lysis buffer followed by RNA extraction using the Qiagen RNeasy Kit, RT-PCR, and Affymetrix microarray analysis of RNA expression.

**Fluorescence in situ hybridization and immunocytochemistry**. For double labeling of lncNB1 RNA and RPL35 protein by fluorescence in situ hybridization and immunocytochemistry, BE(2)-C cells were seeded into cell culture media in eight-well Lab-tek chamber slides (Thermo Fisher Scientific). After fixation in 4% paraformaldehyde at room temperature for 30 min, the cells were treated with 3% hydrogen peroxide for 10 min at room temperature and washed in distilled water twice and PBS once. FISH was performed through probe hybridization and amplification with the RNAscope Multiple Fluorescent Reagent Kit v2 Assay (catalog number 323100-USM; Advanced Cell Diagnostics, Newark, CA). The cells were hybridized with DapB negative control or custom-made lncNB1 RNA probes (catalog numbers 310043 and 577481, respectively; Advanced Cell Diagnostics) for 2 h at 40 °C, followed by three amplification steps. The probes were developed by incubating the cells with RNAscope Multiplex FL v2 HRP-C1 for 15 min at 40 °C, and Opal™ 690 (PerkinElmer, Waltham, MA) diluted in 1× Plus Amplification Diluent (1:150) (PerkinElmer) for 30 min at 40 °C and washed with 1× wash buffer for 2 min at room temperature. The cells were then incubated with RNAscope® Multiplex FL v2 HRP blocker for 15 min at 40 °C and washed.

After the FISH experiment, the cells were blocked with 10% fetal calf serum and 1% bovine serum albumin for 30 min at room temperature, followed by incubation with a control IgG or rabbit anti-RPL35 primary antibody (ab209087; Abcam, Cambridge, UK) (1:30) in 10% fetal calf serum for 1.5 h at room temperature. After washing, the cells were incubated with a donkey anti-rabbit antibody (GEHENA934-1ML; GE Healthcare, Chicago, IL) diluted (1:1000) in 10% fetal calf serum for 1.5 h. The cells were then washed and incubated with Opal™ 520 (PerkinElmer) diluted in 1× Plus Amplification Diluent (1:150) for 10 min at room temperature. Nuclei were counterstained with 4′,6-diamidino-2-phenylindole (DAPI) and the slides were mounted with Vectashield antifade mounting medium (Vector Laboratories, Peterborough, UK). Cell images were captured under a confocal fluorescence microscope (Leica TCS SP8, Wetzlar, Germany).

**Alamar blue assays**. Alamar blue assays were performed with our established protocol[39]. Briefly, cells were transfected with various siRNAs or treated with vehicle control or doxycycline in 96-well plates. After 96 h, the cells were incubated with Alamar blue (Invitrogen) for the last 5 h, and the plates were read on a microplate reader at 570/595 nm. Results were calculated according to the optical density absorbance units and expressed as percentage changes in the number of viable cells, relative to control siRNA-transfected or vehicle control-treated samples.

**BrdU incorporation assays**. Neuroblastoma cells were transfected with control siRNA, lncNB1 siRNA-1 or lncNB1 siRNA-2, or treated with vehicle control or doxycycline in 96-well plates for 96 h. The cells were then incubated with 5-bromo-2′-deoxyuridine (BrdU) (Roche) for the final 2 h, fixed, and incubated with peroxidase-conjugated anti-BrdU antibody. After washing, the peroxidase substrate containing tetramethyl-benzidine (TMB) (Roche) was added, and the plates were read on a microplate reader at 405 nm test wavelength and 415 nm reference wavelength. Results were calculated according to the optical density absorbance units and expressed as percentage changes in BrdU incorporation, relative to control siRNA-transfected or vehicle control-treated samples.

**Cell cycle analysis**. Neuroblastoma cells were transfected with control siRNA, lncNB1 siRNA-1 or lncNB1 siRNA-2, or treated with vehicle control or doxycycline for 96 h. Cells were then harvested, fixed, and resuspended at a concentration of $2 \times 10^6$ cells per ml in solution containing 2 μg per ml RNase (Sigma) and 50 μg per ml propidium iodide (Sigma). The cells were examined by flow cytometry using FACS Calibur machine and FACS Diva software (BD Biosciences). The percentage of cells at each phase of the cell cycle was analyzed using FlowJo Version 10 (TreeStar Inc., Ashland, OR)[39].

**Apoptosis analysis**. Neuroblastoma cells were transfected with control siRNA, lncNB1 siRNA-1 or lncNB1 siRNA-2, or treated with vehicle control or

doxycycline for 96 h. Cells were then stained with Annexin V-FITC and propidium iodide, followed by flow cytometry analysis of cells positively stained by Annexin V-FITC and/or propidium iodide, using FACS Canto Flow Cytometer (BD Biosciences). The percentage of Annexin V and/or propidium iodide positive cells was analyzed with FlowJo Version 10 (ref. [47]).

**In vivo mouse experiments**. Animal experiments were approved by the Animal Care and Ethics Committee of University of New South Wales, Australia, and the animals were cared for in agreement with institutional ethics guidelines. Female Balb/c nude mice aged 5–6 weeks were anesthetized and injected subcutaneously with either $2 \times 10^6$ doxycycline-inducible lncNB1 shRNA-1 BE(2)-C cells or $8 \times 10^6$ doxycycline-inducible lncNB1 shRNA-2 Kelly cells into the right flank. Mice were monitored for tumor development and tumor volume calculated using (length × width × height)/2. Mice were fed with either control food without doxycycline or food containing doxycycline at 600 mg per kg (Meat Free Rat and Mouse ± 600 mg doxycycline per kg Finished Diet; Specialty Feeds Pty Ltd, Glen Forrest, WA, Australia), when tumors reached $0.050 \, cm^3$ in volume. Tumor sizes were measure once every 2 days. Mice were culled when tumors reached $1.0 \, cm^3$.

**Patient tumor sample analysis**. lncNB1, DEPDC1B, E2F1, and RPL35 RNA expression was extracted from the publicly available RNA sequencing SEQC-RPM-seqcnb1 dataset consisting of 493 human neuroblastoma samples with detailed information on MYCN amplification status and clinical outcome [http://r2.amc.nl][26].

**Statistical analysis**. Experiments for statistical analysis were performed at least three times. Data were examined with Graphpad Prism 6 program and expressed as mean ± standard error. Differences were analyzed for significance with two-sided unpaired $t$-test for two groups or *ANOVA* among groups.

Correlation between lncNB1, E2F1, and RPL35 expression with DEPDC1B expression in human neuroblastoma tissues was examined using Pearson's correlation. Overall survival of patients was defined as the time from diagnosis until death or until last contact if the patient did not die. Survival analyses were performed using GraphPad Prism 6.0 according to the method of Kaplan and Meier, and comparisons of survival curves were performed using two-sided log-rank tests. Probabilities of survival and hazard ratios (HRs) were provided with 95% confidence intervals (CIs). Proportionality was confirmed by visual inspection of the plots of $\log(2 \log(S(\text{time})))$ versus $\log(\text{time})$, which were found to remain parallel[8]. A probability value of 0.05 or less was considered statistically significant. All statistical tests were two-sided.

## Data availability

The RNA sequencing, differential gene expression microarray data, and the differential gene expression microarray data from polysome profiling have been deposited at the Gene Expression Omnibus website with series numbers of GSE114053, GSE113998, and GSE132109 respectively. Gene expression in the publicly available RNA sequencing SEQC-RPM-seqcnb1 dataset consisting of 493 human neuroblastoma samples were downloaded from the R2 platform [http://r2.amc.nl]. All other relevant data are available from the corresponding authors on request. The source data underlying Figs. 1c-f, 2b-e, 3a-e, 4a-e, 5a-g, 6a-g, 7a-d, 8a-l and Supplementary Figs 1b, c, 2a-c, 3a-f, 4a-e, 5b, c, 6a-d, 8 and 9a-e, uncropped and unprocessed immunoblot scans are provided as a Source Data file. Gene expression in 493 human neuroblastoma samples and relevant patient prognosis information in the Tumor Neuroblastoma SEQC-RPM-seqcnb1 dataset, as well as gene expression in 88 human neuroblastoma samples and relevant patient prognosis information in the Tumor Neuroblastoma public-Versteeg dataset, were downloaded from the R2 platform [http://r2.amc.nl].

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

## Acknowledgements

We thank Dr. Marco Herold at Walter and Eliza Hall Institute of Medical Research, Melbourne, Australia, for providing the FH1tUTG construct. Children's Cancer Institute Australia is affiliated with UNSW Australia and Sydney Children's Hospitals Network. The authors were supported by National Health & Medical Research Council Australia, National Institutes of Health USA (CA226959-01), Italian Association for Research on Cancer (AIRC), and Cancer Council New South Wales. P.Y.L. is a research fellow of Cancer Institute New South Wales.

## Author contributions

P.Y.L., A.E.T., G.M., K.M.H., J.M., S.M., B.A., N.B., H.P., N.H., C.M., R.C., Y.S., M.J.H., J.G., C.E., A.J.H., M.W., L.S., J.V., J.L., and P.M. performed the experiments, collected the data, and analyzed the results. B.B.C., T.S., M.F., G.M.M., M.D.N., and M.H. contributed biological samples and clinical data. Q.L., J.Y.W., X.D.Z, R.D.H., and M.E.D. provided conceptual advice. T.L. and G.P. designed and supervised the study. P.Y.L., A.E.T., G.M., J.M., M.E.D., G.P., and T.L. wrote the manuscript with contributions from the co-authors.

## Competing interests

The authors declare no competing interests.
