## [Peer Review File · Nature Communications]

Reviewers' comments:

Reviewer #1 (Remarks to the Author):

In the manuscript entitled "The novel long noncoding RNA lncNB1 promotes tumorigenesis by interacting with ribosomal protein RPL35" by Liu et al, the authors present a novel lncRNA, lncNB1, overexpressed in MYCN amplified neuroblastoma, which mechanistically binds the ribosomal protein RPL35, inducing the stabilization of N-MYC. The research is well conducted and results are displayed in a logical manner, but there are several concerns which need to be answered:

Results:

The novel lncRNA, lncNB1, is over-expressed in MYCN-amplified human neuroblastoma tissues and cells.

1) Except MYCN, MYCNOS, IGF2BP1 and lncNB1, there are other 4 RNAs specifically overexpressed in MYCN-amplified cells. The name should be added to the main figure 1b and also their role should be discussed, are these molecules reported in the literature to associate with N-MYC neuroblastoma in cancers?

2) Is lncNB1 poly adenylated?

3) Seeing that you used a public available SNP array data - are there any SNPs present in the genome of the lncNB1?

4) The characterization of lncNB1 is incomplete – add data on the abundance of the lncNB1, add data on the conservation between species of the lncNB1, add data on the genomic localization of lncNB1 and the data on the cellular localization of the lncRNA should be moved to the first chapter.

lncNB1 is required for DEPDC1B mRNA and protein and E2F1 protein expression.

1) Why did you transfect the cells for 40 hours in order to check for the function of the lncRNA?

2) Add an image/table of the results of the affymetrix microarray? What other genes were deregulated except DEPDC1B?

3) Fig 2a: the p value and the FDR are 0.00, add the numbers.

4) The sequence in which the samples are loaded in the immunoblot is very peculiar: siRNA 1, control, siRNA-2; and is different from the sequence of the RT-PCR. Why were the samples loaded in this order?

lncNB1 up-regulates N-Myc protein expression through DEPDC1B-mediated ERK protein phosphorylation and N-Myc protein stabilization.

1. The overexpression experiments with lncNB1 and DEPDC1 are essential and should be performed in a second cell line, not only BE(2)-C.

2. Figure 3c – it would be more powerful data and similar to the pathogenic reality if you would perform the chase assay in stable clones with overexpression of lncNB1 vs empty.

lncNB1 activates DEPDC1B gene transcription through the transcription factor E2F1.

1. Add some imaging studies (FISH) to show that your lncRNA is located mostly in the cytoplasm.

lncNB1 up-regulates E2F1 expression by binding to the ribosomal protein RPL35 and increasing E2F1 protein synthesis.

1. You mention that your lncRNA is located in the cytoplasm add FISH imaging data to show the colocalization of lncNB1 with RPL35.

2. RPL35 is a ribosomal protein which could affect the translation of very many proteins, how specific does the interaction lncNB1 – RPL35 affect only the level of E2F1? How many other proteins are altered? Clearly actin is not affected. And if only E2F1 is affected, how can be the translation be so specific?

Methods:

1. RNA sequencing data analysis – References are not introduced correctly.

2. qRT-PCR for patient samples - it is recommended to use two different normalizers which are

stable between the groups.

Reviewer #2 (Remarks to the Author):

Review of NCOMMS-18-34609

The novel long noncoding RNA lncNB1 promotes tumorigenesis by interacting with ribosomal protein RPL35

In this ms. Liu et al. describe the identification of a novel long noncoding RNA lncNB1. They perform functional studies using siRNA and shRNA-mediated knockdown of lncNB1 and identify that lncNB1 is essential for proliferation and survival of neuroblastoma cells.

Major

1. In figure 1, the authors use a very selected set of neuroblastoma cells lines with and without amplification of the MYCN oncogene to derive a set of 459 genes associated with MYCN amplification. Given the independent dataset (39 neuroblastoma cell lines – Maris dataset) in figure 1e, it appears that the correlation of lncNB1 and MYCN is not that strict as suggested by the limited number of cell lines analyzed in figure 1a. For instance, figure 1e shows four cell lines with high mRNA expression of MYCN that do not express lncNB1. One of these cell lines is SK-N-BE2, a cell line very closely related to SK-N-BE2(c). In contrast to SK-N-BE2(c), SK-N-BE2 does not express lncNB1, but has MYCN amplification and expression (in 39 cell line dataset – Maris), suggesting that lncNB1 is not required in cell lines with MYCN amplification. Furthermore, figure 1d suggests that cell lines without amplification of MYCN do not express lncNB1 RNA, while it appears that in figure 1e lncNB1 is expressed in cell lines with very low expression of MYCN. Therefore, the correlation of lncNB1 and MYCN is more complex than suggested by the authors and does not support the author's claim that '...lncNB1...correlates with MYCN gene amplification and expression.'

2. The authors show in figure 6 that knockdown of lncNB1 for 96hrs shows a very strong reduction in cell viability and an induction of Annexin V-positive apoptotic cells. Importantly, the apoptotic effect of lncNB1 silencing is that strong that none of the cells with knockdown of lncNB1 generates colonies, as shown in figure 7a. The same cells with inducible shRNA of lncNB1 are used in figures 2d,e and 3e to identify E2F1 and MYCN as target genes at 48 hours after silencing lncNB1. The fact the lncNB1 leads to a reduction of cell proliferation and an induction of apoptosis is a very likely explanation for the observed changes in E2F1 and MYCN, as their expression is tightly connected to the cell cycle. Importantly, induction of apoptosis leads to strong alterations in the regulation of many genes, doubting the specificity of the regulatory relationship of lncNB1 and MYCN.

3. The authors claim a regulatory cascade stating that 'lncNB1 regulates MYCN protein through DEPDC1B-mediated ERK protein phosphorylation', is not proven. Instead, the authors have performed manipulation of several genes like DEPDC1B that all reduce cell viability. As a consequence of the reduction in cell viability, such analyses may lead to a correlative change in downstream genes, which is not necessarily the same as a cascade of these genes. Without a rescue experiment with a downstream gene and restoration of target gene expression, proof for a cascade is lacking.

4. The authors claim from the Kaplan-Meier survival analyses that lncNB1 predicts a poor prognosis of neuroblastoma patients. This is not the case, since expression of lncNB1 is higher in INSS stage 4 neuroblastoma tumors. INSS stage 4 tumors have a very poor prognosis. Neuroblastoma has many INSS stages with a good prognosis (i.e. INSS stage 1, 2 and 4S), which are all included in these analyses. By definition, in a survival analysis that includes neuroblastoma tumors from INSS stage 1, 2, 3, 4 and 4s, a gene that is higher expressed in INSS stage 4 will

associate with a poor prognosis. Limiting the analysis to INSS stage 4 neuroblastoma only, the relation between higher expression of lncNB1 and patient prognosis is lost. The author's conclusion that high levels of lncNB1 predict poor prognosis is therefore of very limited value, since it is not independent from other variables used in the clinical staging of neuroblastoma (e.g. INSS).

Tao Liu, BMed, MMed, PhD
Group Leader, Histone Modification Group
Children's Cancer Institute Australia
Lowy Cancer Research Centre, UNSW
Randwick, Sydney, NSW 2031
Australia
Ph: 61 2 9385 1935
E-mail: tliu@ccia.unsw.edu.au

July 30, 2019

Reviewers
Nature Communications

Re: Submission of revised manuscript NCOMMS-18-34609 entitled “*The novel long noncoding RNA lncNB1 promotes tumorigenesis by interacting with ribosomal protein*”

Dear Reviewers,

Thank you for reviewing the manuscript and for the opportunity to submit a revision. We appreciate the favourable evaluations and constructive comments. We have added substantial new data to results, figures and supplementary information figures and tables, and modified the manuscript to address each of the points raised, as detailed below in “Responses to Reviewers” and as indicated in the article file.

Responses to Reviewers

Reviewer #1:

Question 1 (1-4). The novel lncRNA, lncNB1, is over-expressed in MYCN-amplified human neuroblastoma tissues and cells.

1(1). Except MYCN, MYCNOS, IGF2BP1 and lncNB1, there are other 4 RNAs specifically overexpressed in MYCN-amplified cells. The name should be added to the main figure 1b and also their role should be discussed, are these molecules reported in the literature to associate with N-MYC neuroblastoma in cancers?

Responses:

We have added the names of the other 4 RNAs, RNF217, GRIK3, SLCO5A1 and RP11-102F4.3, which are considerably over-expressed in MYCN-amplified neuroblastoma cell lines, into Fig. 1b. RNF217, GRIK3 and SLCO5A1 promotes leukemia, breast and colon cancer respectively, but RP11-102F4.3 has never been studied in normal physiology or disease. We have also added information about the 4 RNAs into the “Results” section and discussed their functional roles in the “Discussion” section.

Changes:

- We have added the names of the other 4 RNAs, RNF217, GRIK3, SLCO5A1 and RP11-102F4.3, which are considerably over-expressed in MYCN-amplified neuroblastoma cell lines, into Fig. 1b.
- We have added the following sentence from the 14th to the 15th line on page 5 in the “**Results**” section: “.....The other RNAs considerably over-expressed in MYCN-amplified neuroblastoma cell lines were RP11-102F4.3, RNF217, GRIK3 and SLCO5A1 (Fig. 1a,b).....”
- We have added the following sentences from the 4th to the 10th line on page 15 in the “**Discussion**” section: “.....Here our RNA sequencing analysis identifies N-Myc, MYCNOS, IGF2BP1, lncNB1, RNF217, RP11-102F4.3, GRIK3 and SLCO5A1 as the RNAs most considerably over-expressed in MYCN-amplified, compared with MYCN-non-amplified, neuroblastoma cell lines. MYCNOS¹⁸ and IGF2BP1¹⁶ promote MYCN-driven neuroblastoma metastasis and tumorigenesis respectively, RNF217²⁷, GRIK3²⁸ and SLCO5A1²⁹ promotes leukemia, breast and colon cancer. However, the lncRNAs lncNB1 and RP11-102F4.3 have never been studied in normal physiology or disease.....”

Question 1(2). Is lncNB1 poly adenylated?

Response:

Yes, lncNB1 is poly adenylated.

Change:

- We have revised the following sentence from the 10th to the 14th line on page 5 in the “**Results**” section: “.....Interestingly, the lncRNA RP1-40E16.9 at chromosome 6: 3182817-3195767, also known as LOC100507194 and LINC02525 but will herein be referred to as lncNB1 (lncRNA highly expressed in neuroblastoma 1), was polyadenylated and displayed an expression pattern very similar to N-Myc and MYCNOS (Fig. 1a,b), suggesting that this lncRNA could be involved in MYCN-amplified neuroblastoma tumorigenesis.....”

Question 1(3). Seeing that you used a public available SNP array data - are there any SNPs present in the genome of the lncNB1?

Response:

The SNP array data did not reveal SNP at the lncNB1 gene.

Question 1(4). The characterization of lncNB1 is incomplete - add data on the abundance of the lncNB1, add data on the conservation between species of the lncNB1, add data on the genomic localization of lncNB1 and the data on the cellular localization of the lncRNA should be moved to the first chapter.

Responses:

We have examined the abundance of lncNB1 transcript among various human tissues, using the publicly available Genotype-Tissue Expression (GTEx) Release V7 dataset (dbGaP Accession phs000424.v7.p2). The GTEx Release V7 dataset provides RNA sequencing transcript expression data from 53 non-diseased tissues of different sites across nearly 1000 people. Analysis of the GTEx Release V7 dataset showed that lncNB1 is expressed in brain, pituitary, testis, uterus and nerve tissues, but hardly detectable in other human tissues (new Supplementary Fig. 1a).

Analyses with the PhastCons program show that lncNB1 is not conserved between homo sapiens and mouse, zebrafish or fruit flies. The poor conservation of lncNB1 may be explained by its highest expression levels in brain tissues.

We have added genomic localization of lncNB1 (chromosome 6: 3182817-3195767) into the “Results” section.

We have moved the data on lncNB1 RNA cellular localization to the first part of the “Results” section, as the new Supplementary Figures 1b and 1c, as suggested by the reviewer.

Changes:

- We have added the new data on lncNB1 transcript abundance in normal human tissues as the new Supplementary Fig. 1a, and moved the data on lncNB1 RNA cellular localization to the first part of the “**Results**” section as the new Supplementary Fig. 1b,c.
- We have revised the following sentence from the 10th to the 12th line on page 5 in the “**Results**” section: “.....Interestingly, the lncRNA RP1-40E16.9 at chromosome 6: 3182817-3195767, also known as LOC100507194 and LINC02525 but will herein be referred to as lncNB1 (lncRNA highly expressed in neuroblastoma 1).....”
- We have revised the following sentences from the 16th to the 20th line on page 5 in the “**Results**” section: “.....The publicly available Genotype-Tissue Expression (GTEx) Release V7 dataset (dbGaP Accession phs000424.v7.p2) provides RNA sequencing transcript expression data from 53 non-diseased tissues of different sites across nearly 1000 people. Analysis of the GTEx Release V7 dataset showed that lncNB1 was expressed in brain, pituitary, testis, uterus and nerve tissues, but hardly detectable in other human tissues (Supplementary Fig. 1a).....”
- We have revised the following sentence from the 12th to the 13th line on page 6 in the “**Results**” section: “.....In addition, RT-PCR analysis showed that lncNB1 RNA was mainly localized in the cytoplasm not the nucleus (Supplementary Fig. 1b,c).....”

Question 2 (1-4). lncNB1 is required for DEPDC1B mRNA and protein and E2F1 protein expression.

Question 2(1) Why did you transfect the cells for 40 hours in order to check for the function of the lncRNA?

Response:

To avoid gene expression changes due to neuroblastoma cell growth inhibition or apoptosis, we performed microarray experiments in neuroblastoma cells 40 hours after transfection with control siRNA or lncNB1 siRNAs. Our cell cycle, cell proliferation and apoptosis assays have confirmed that lncNB1 knockdown for 48 hours do not have an effect on neuroblastoma cell proliferation or survival (new Supplementary Fig. 4), and that lncNB1 knockdown for 96 hours reduces neuroblastoma cell proliferation and survival (Fig. 6a,b,f,g, and new Supplementary Fig. 8).

Question 2(2). Add an image/table of the results of the affymetrix microarray? What other genes were deregulated except DEPDC1B?

Response and Changes:

We have added the Affymetrix microarray data as the new Supplementary Table 3.

Question 2(3). Fig 2a: the p value and the FDR are 0.00, add the numbers.

Response and Changes:

We have added the p value and the FDR value to Fig. 2a.

Question 2(4). The sequence in which the samples are loaded in the immunoblot is very peculiar: siRNA 1, control, siRNA-2; and is different

from the sequence of the RT-PCR. Why were the samples loaded in this order?

Response:

For immunoblot, we always load protein from control siRNA-transfected cells in the middle, and protein from target siRNA-transfected samples on the two sides. For PRISM histograms of RT-PCR data, we always list data from control siRNA-transfected cells first, followed by data from target siRNA-transfected cells.

Question 3 (1-2). lncNB1 up-regulates N-Myc protein expression through DEPDC1B-mediated ERK protein phosphorylation and N-Myc protein stabilization.

Question 3(1). The overexpression experiments with lncNB1 and DEPDC1 are essential and should be performed in a second cell line, not only BE(2)-C.

Response:

In the original manuscript, we showed that transfection with a lncNB1 or DEPDC1B expression construct for 48 hours led to DEPDC1B over-expression, enhanced ERK protein phosphorylation and N-Myc protein over-expression in BE(2)-C neuroblastoma cells (original Supplementary Fig. 3d).

To address this question, we have transfected Kelly neuroblastoma cells with an empty vector, lncNB1 or DEPDC1B expression construct for 48 hours. Immunoblot analysis showed that forced lncNB1 or DEPDC1B over-expression in Kelly cells also resulted in DEPDC1B over-expression, ERK protein phosphorylation and N-Myc protein over-expression (new Supplementary Fig. 3d). In addition, DOX-inducible lncNB1 shRNA-1 BE(2)-C and Kelly cells were transfected with an empty vector or DEPDC1B expression construct and treated with vehicle control or DOX for 48 hours. Immunoblot analysis showed that forced DEPDC1B over-expression largely reversed lncNB1 shRNA-mediated ERK protein de-phosphorylation, N-Myc protein de-phosphorylation at S61 and N-Myc protein reduction (new Supplementary Fig. 3e). Taken together, the data suggest that lncNB1 and DEPDC1B up-regulates N-Myc protein expression.

Changes:

- We have added the new immunoblot data into the new Supplementary Fig. 3d,e.
- We have revised the following sentence from the 2nd line to the 9th line on page 8 in the Results section: “.....In addition, transfection with a lncNB1 or DEPDC1B over-expression construct in BE(2)-C and Kelly cells up-regulated DEPDC1B protein expression, ERK protein phosphorylation as well as N-Myc protein expression (Supplementary Fig. 3d). DOX-inducible lncNB1 shRNA BE(2)-C and Kelly cells were then transfected with an empty vector or DEPDC1B expression construct and treated with vehicle control or DOX for 48 hours. Immunoblot analysis showed that forced DEPDC1B over-expression largely reversed lncNB1 shRNA-mediated ERK protein de-phosphorylation, N-Myc protein de-phosphorylation at S62 and N-Myc protein reduction (Supplementary Fig. 3e).....”

Question 3(2). Figure 3c – it would be more powerful data and similar to the pathogenic reality if you would perform the chase assay in stable clones with overexpression of lncNB1 vs empty.

Response:

To address this question, we have transfected BE(2)-C neuroblastoma cells with an empty vector or lncNB1 expression construct for 48 hours. Cells were treated with vehicle control or cycloheximide for the last 0, 15, 30, 45 or 60 minutes. Immunoblot analysis showed that forced lncNB1 over-expression increased the half-life of N-Myc protein from 43 minutes in cells transfected with an empty vector to 60 minutes in cells transfected with a lncNB1 expression construct (new Supplementary Fig. 3f).

Changes:

- We have added the new pulse chase assay data as the new Supplementary Fig. 3f.
- We have revised the following sentences from the 12th line to the 16th line on page 8 in the Results section: “.....In addition, BE(2)-C cells were transfected with an empty vector or lncNB1 expression construct, followed by treatment with vehicle control or cycloheximide. The pulse chase assays showed that N-Myc protein half-life was reduced by approximately 50% by lncNB1 siRNA or DEPDC1B siRNA (Fig. 3c), and was increased by approximately 39% by the lncNB1 expression construct (Supplementary Fig. 3f).”

Question 4. lncNB1 activates DEPDC1B gene transcription through the transcription factor E2F1.

Add some imaging studies (FISH) to show that your lncRNA is located mostly in the cytoplasm.

Responses:

To address this question, we have performed double labelling experiments with fluorescence in situ hybridization (FISH) with probes targeting a negative control or lncNB1 RNA and immunocytochemistry with a control IgG or anti-RPL35 antibody in neuroblastoma cells. The double labelling experiments showed that lncNB1 RNA and RPL35 protein are both mostly located in the cytoplasm (new Supplementary Fig. 7a,b).

Changes:

- We have added the double labelling FISH and immunocytochemistry data as the new Supplementary Fig. 7a,b.
- We have revised the following sentence from the 1st line to the 3rd line on page 12 in the “Results” section: “.....fluorescence in situ hybridization and immunocytochemistry double labelling experiments showed that lncNB1 RNA and RPL35 protein were mostly located in the cytoplasm in neuroblastoma cells (Supplementary Fig. 7a,b).....”.
- We have also added the following two paragraphs on page 28 and page 29 in the “Methods” section:

Fluorescence in situ hybridization and immunocytochemistry. For double labelling of lncNB1 RNA and RPL35 protein by fluorescence in situ hybridization and immunocytochemistry, BE(2)-C cells were seeded into cell culture media in eight-well Lab-tek chamber slides (Thermo Fisher Scientific). After fixation in 4% paraformaldehyde at room temperature for 30 minutes, the cells were treated with 3% hydrogen peroxide for 10 minutes at room temperature and washed in distilled water twice and PBS once. FISH was performed through probe hybridization and amplification with the RNAscope Multiple Fluorescent Reagent Kit v2 Assay (catalogue number 323100-USM, Advanced Cell Diagnostics, Newark, CA, USA). The cells were hybridized with DapB negative control or custom-made lncNB1 RNA probes (catalogue numbers 310043 and 577481 respectively, Advanced Cell Diagnostics) for 2 hours at 40°C, followed by three amplification steps. The probes were developed by incubating the cells with RNAscope Multiplex FL v2 HRP-C1 for 15 minutes at 40°C, and Opal™ 690 (PerkinElmer, Waltham, MA, USA) diluted in 1 × Plus Amplification Diluent

(1:150) (PerkinElmer) for 30 minutes at 40°C and washed with 1 × wash buffer for 2 minutes at room temperature. The cells were then incubated with RNAscope® Multiplex FL v2 HRP blocker for 15 minutes at 40°C and washed.

After the FISH experiment, the cells were blocked with 10% fetal calf serum and 1% bovine serum albumin for 30 minutes at room temperature, followed by incubation with a control IgG or rabbit anti-RPL35 primary antibody (Abcam, Cambridge, UK) (1:30) in 10% fetal calf serum for 1.5 hours at room temperature. After washing, the cells were incubated with a donkey anti-rabbit antibody (GE Healthcare, Chicago, IL, USA) diluted (1:1000) in 10% fetal calf serum for 1.5 hours. The cells were then washed and incubated with Opal™ 520 (PerkinElmer) diluted in 1 × Plus Amplification Diluent (1:150) for 10 minutes at room temperature. Nuclei were counterstained with 4',6-diamidino-2-phenylindole (DAPI) and the slides were mounted with Vectashield antifade mounting medium (Vector Laboratories, Peterborough, UK). Cell images were captured under a confocal fluorescence microscope (Leica TCS SP8, Wetzlar, Germany).

Question 5. lncNB1 up-regulates E2F1 expression by binding to the ribosomal protein RPL35 and increasing E2F1 protein synthesis.

Question 5(1). You mention that your lncRNA is located in the cytoplasm add FISH imaging data to show the colocalization of lncNB1 with RPL35.

Responses:

To address this question, we have performed double labelling experiments with fluorescence in situ hybridization (FISH) with probes targeting a negative control or lncNB1 RNA and immunocytochemistry with a control IgG or anti-RPL35 antibody in neuroblastoma cells. The double labelling experiments showed that lncNB1 RNA and RPL35 protein are both mostly located in the cytoplasm (new Supplementary Fig. 7a,b).

Changes:

Please refer to Changes for Question 4.

Question 5(2). RPL35 is a ribosomal protein which could affect the translation of very many proteins, how specific does the interaction lncNB1 - RPL35 affect only the level of E2F1? How many other proteins are altered? Clearly actin is not affected. And if only E2F1 is affected, how can be the translation be so specific?

Responses:

To address this question, we have transfected BE(2)-C neuroblastoma cells with control siRNA, lncNB1 siRNA-1 or lncNB1 siRNA-2 for 48 hours, followed by polysome fractionation. Affymetrix microarray experiments were performed with RNA samples from the heavy polysomes. Differential gene expression analysis showed that lncNB1 siRNA-1 and lncNB1 siRNA-2 commonly reduced the expression of 34 mRNAs, including E2F1 mRNA, in the heavy polysomes (new Supplementary Table 6). The data confirm that lncNB1 induces the translation of mRNAs, including E2F1 mRNA, into proteins.

Changes:

- We have added the new Affymetrix microarray data on mRNAs down-regulated in heavy polysomes in BE(2)-C cells after lncNB1 knockdown as the new Supplementary Table 6.
- We have added the following sentence from the second last line on page 11 to the 1st line on page 12 in the “**Results**” section: “.....In addition, Affymetrix microarray experiments

revealed that lncNB1 knockdown resulted in reduction in the expression of 34 mRNAs, including E2F1 mRNA, in the heavy polysomes (Supplementary Table 6).”

Question 6 (1-2). Methods:

Question 6 (1). RNA sequencing data analysis – References are not introduced correctly.

Response and change.

We have revised the references for RNA sequencing data analysis in the “**Methods**” section.

Question 6 (2). qRT-PCR for patient samples - it is recommended to use two different normalizers which are stable between the groups.

Response:

For patient sample analysis, we used the publicly available SEQC-RPM-seqcnb1 human neuroblastoma tissue RNA sequencing-patient prognosis dataset downloaded from the R2 platform (<http://r2.amc.nl>), not qRT-PCR.

REVIEWER #2

Question 1. In figure 1, the authors use a very selected set of neuroblastoma cell lines with and without amplification of the MYCN oncogene to derive a set of 459 genes associated with MYCN amplification. Given the independent dataset (39 neuroblastoma cell lines – Maris dataset) in figure 1e, it appears that the correlation of lncNB1 and MYCN is not that strict as suggested by the limited number of cell lines analyzed in figure 1a. For instance, figure 1e shows four cell lines with high mRNA expression of MYCN that do not express lncNB1. One of these cell lines is SK-N-BE2, a cell line very closely related to SK-N-BE2(c). In contrast to SK-N-BE2(c), SK-N-BE2 does not express lncNB1, but has MYCN amplification and expression (in 39 cell line dataset – Maris), suggesting that lncNB1 is not required in cell lines with MYCN amplification. Furthermore, figure 1d suggests that cell lines without amplification of MYCN do not express lncNB1 RNA, while it appears that in figure 1e lncNB1 is expressed in cell lines with very low expression of MYCN. Therefore, the correlation of lncNB1 and MYCN is more complex than suggested by the authors and does not support the author’s claim that ‘...lncNB1...correlates with MYCN gene amplification and expression.’

Response.

We agree with the Reviewer that few MYCN-amplified human neuroblastoma cell lines express low levels of lncNB1, and few MYCN-non-amplified neuroblastoma cell lines express high levels of lncNB1, in the 39 human neuroblastoma cell line dataset (Maris - 41 - FPKM - rsg001 RNA sequencing dataset). This can be explained by the heterogeneity of neuroblastoma, since human neuroblastoma cell lines and tissues harbour various genetic abnormalities, such as whole or segmental chromosome gain/deletion and chromosomal rearrangement involving transcriptional enhancers (such as TERT gene rearrangement with transcriptional super-enhancers, c-Myc gene focal enhancer amplification and genomic rearrangements and consequent enhancer hijacking). It is

reasonable to hypothesize that lncNB1 gene expression is likely to correlate with multiple factors, including MYCN gene amplification, chromosomal deletion, gain and rearrangements.

Importantly, as shown in Figures 1e and 1f, in the same 39 human neuroblastoma cell lines (Maris - 41 - FPKM - rsg001 RNA sequencing dataset), lncNB1 gene expression positively correlates with MYCN gene expression ($R = 0.503$, $p = 0.001$, Figure 1e), and lncNB1 gene expression was significantly higher in MYCN gene-amplified than MYCN-non-amplified human neuroblastoma cell lines ($p = 0.001$, Figure 1f). To further address the question, we examined lncNB1 expression in 493 human neuroblastoma tissues in the publicly available SEQC-RPM-seqcnb1 human neuroblastoma tissue RNA sequencing dataset. Five out of 498 samples were excluded from the analysis, due to the lack of information on MYCN gene amplification status. Two-sided Pearson's correlation study and Student's t-test showed that lncNB1 RNA expression positively correlated with MYCN gene amplification ($p < 0.001$, original Fig. 8a) and N-Myc mRNA expression ($p < 0.001$, new Supplementary Figure 9a). Taken together, we can safely conclude that lncNB1 is over-expressed in MYCN gene-amplified human neuroblastoma cell lines and tissues, and that lncNB1 over-expression correlates with MYCN gene amplification and over-expression.

Changes:

- We have added the new Pearson's correlation study of the correlation between lncNB1 RNA expression and N-Myc mRNA expression in 493 human neuroblastoma tissues as the new Supplementary Figure 9a).
- We have added the following sentence from the 2nd last line on page 13 to the 4th line on page 14 in the "**Results**" section: ".....Two-sided Pearson's correlation study and Student's t-test showed that lncNB1 RNA expression positively correlated with MYCN amplification (Fig. 8a), N-Myc mRNA expression (Supplementary Fig. 9a) as well as DEPDC1B mRNA expression (Fig. 8b), and that RPL35 and E2F1 mRNA expression also positively correlated with DEPDC1B mRNA expression in the 493 human neuroblastoma tissues (Fig. 8c,d)....."

Question 2. The authors show in figure 6 that knockdown of lncNB1 for 96hrs shows a very strong reduction in cell viability and an induction of Annexin V-positive apoptotic cells. Importantly, the apoptotic effect of lncNB1 silencing is that strong that none of the cells with knockdown of lncNB1 generates colonies, as shown in figure 7a. The same cells with inducible shRNA of lncNB1 are used in figures 2d,e and 3e to identify E2F1 and MYCN as target genes at 48 hours after silencing lncNB1. The fact the lncNB1 leads to a reduction of cell proliferation and an induction of apoptosis is a very likely explanation for the observed changes in E2F1 and MYCN, as their expression is tightly connected to the cell cycle. Importantly, induction of apoptosis leads to strong alterations in the regulation of many genes, doubting the specificity of the regulatory relationship of lncNB1 and MYCN.

Responses:

In the original manuscript, we have found that knocking down lncNB1 for 96 hours considerably reduces neuroblastoma cell proliferation and induces apoptosis (Fig. 6a,b,f,g), and that knocking down lncNB1 for two weeks almost completely abolishes neuroblastoma cell clonogenic capacity (Fig. 7a,b).

In the original manuscript, our *microarray experiments identified E2F1 and Myc pathways as the main signalling pathways significantly suppressed, 40 hours after lncNB1 knockdown* (Fig.

2a & Supplementary Table 4). Our RT-PCR and immunoblot experiments confirmed that DEPDC1B protein, DEPDC1B mRNA, E2F1 and N-Myc protein but not mRNA were significantly down-regulated, 48 hours after lncNB1 knockdown (Figures 2b-e and Figures 3a,b,e). Consistent with these data, forced lncNB1 expression for 48 hours enhances DEPDC1B expression, ERK protein phosphorylation and N-Myc protein expression (Supplementary Fig. 3d). Importantly, we have also demonstrated that *lncNB1 siRNA-mediated N-Myc protein reduction is reversed by the proteasome inhibitor MG132 (Fig. 3d)*, and that *N-Myc protein half-life was reduced from 35 minutes to 14 minutes, 30 hours after lncNB1 knockdown (Fig. 3c)*.

To address the reviewer's comments, we have transfected BE(2)-C and Kelly cells with control siRNA, lncNB1 siRNA-1 or lncRNA-2 for 48 hours, followed by incubation with Alamar blue. Alamar blue assays showed that lncNB1 knockdown for 48 hours does not reduce the number of neuroblastoma cells (new Supplementary Fig. 4a). In addition, we have treated DOX-inducible control shRNA, lncNB1 shRNA-1 or lncNB1 shRNA-2 BE(2)-C and Kelly cells with vehicle control or DOX for 48 hours, followed by incubation with Alamar blue or Annexin V. Alamar blue assays and flow cytometry analysis of Annexin V-positively stained cells showed that lncNB1 knockdown for 48 hours does not reduce the number of neuroblastoma cells and does not induce apoptosis (new Supplementary Fig. 4b,c), demonstrating that lncNB1 knockdown for 48 hours is too early to have an effect on cell proliferation or survival.

Taken together, the above data confirm that lncNB1 knockdown reduces E2F1 protein expression and induces N-Myc protein degradation, before cell growth inhibition and apoptosis. We can safely conclude that lncNB1 regulates E2F1 and N-Myc protein expression, not due to its effects on neuroblastoma cell proliferation or survival.

Changes:

- We have added the new Alamar blue assay data from BE(2)-C and Kelly cells after transfection with control siRNA or lncNB1 siRNAs as the new Supplementary Fig. 4a.
- We have added the new Alamar blue assay data and flow cytometry analysis of apoptosis data from DOX-inducible control shRNA, lncNB1 shRNA-1 and lncNB1 shRNA-2 BE(2)-C and Kelly cells after treatment with vehicle control or DOX as the new Supplementary Fig. 4b,c.
- We have added the following sentences as the first paragraph on page 9 in the “**Results**” section: “To exclude cell growth inhibition or cell death as a contributing factor in the regulation of N-Myc expression, we transfected BE(2)-C and Kelly cells with control siRNA, lncNB1 siRNAs or DEPDC1B siRNAs for 48 hours, and treated DOX-inducible control shRNA and lncNB1 shRNA BE(2)-C and Kelly cells with vehicle control or DOX for 48 hours. Alamar blue assays and flow cytometry analyses of Annexin V positively stained cells showed that knocking down lncNB1 or DEPDC1B expression for 48 hours was too early to have an effect on neuroblastoma cell proliferation or survival (Supplementary Fig. 4a-e).....”

Question 3. The authors claim a regulatory cascade stating that ‘lncNB1 regulates MYCN protein through DEPDC1B-mediated ERK protein phosphorylation’, is not proven. Instead, the authors have performed manipulation of several genes like DEPDC1B that all reduce cell viability. As a consequence of the reduction in cell viability, such analyses may lead to a correlative change in downstream genes, which is not necessarily the same as a cascade of these genes. Without a rescue experiment with a downstream gene and restoration of target gene expression, proof for a cascade is lacking.

Responses:

In the original manuscript, our Alamar blue assays showed that knocking down DEPDC1B for 96 hours considerably reduced neuroblastoma cell proliferation (Fig. 6c). Our RT-PCR and immunoblot data confirmed that ERK protein phosphorylation, N-Myc protein phosphorylation at S62 and N-Myc protein but not N-Myc mRNA expression was significantly down-regulated by DEPDC1B siRNAs (Fig. 3b and Supplementary Fig. 3b,c), 48 hours after transfection. Consistent with these data, transfection with a *lncNB1* or DEPDC1B expression construct significantly increased ERK protein phosphorylation and N-Myc protein expression (Supplementary Fig. 3d), 48 hours after transfection. Importantly, *lncNB1* siRNA- and DEPDC1B siRNA-mediated N-Myc protein reduction was reversed by the proteasome inhibitor MG132 (Fig. 3d), and *pulse-chase assays demonstrated that N-Myc protein half-life was reduced by approximately 50%, 30 hours after lncNB1 or DEPDC1B knockdown (Fig. 3c).*

To address the Reviewer's comments, we have transfected BE(2)-C and Kelly neuroblastoma cells with control siRNA, DEPDC1B siRNA-1 or DEPDC1B siRNA-2 for 48 hours, followed by incubation with Alamar blue or Annexin V. Alamar blue assays and flow cytometry analysis of Annexin V-positively stained cells showed that DEPDC1B knockdown for 48 hours did not reduce the number of neuroblastoma cells and did not induce apoptosis (new Supplementary Fig. 4d,e), demonstrating that DEPDC1B knockdown for 48 hours is too early to have an effect on cell proliferation or survival. The data suggest that the reduction in ERK protein phosphorylation, N-Myc protein phosphorylation and expression in neuroblastoma cells 48 hours after DEPDC1B knockdown is not due to cell growth inhibition or cell death.

To further address the Reviewer's comments, we have transfected DOX-inducible *lncNB1* shRNA-1 BE(2)-C and Kelly cells with an empty vector or DEPDC1B expression construct, followed by treatment with vehicle control or DOX for 48 hours. Immunoblot analysis showed that DEPDC1B over-expression increased ERK protein phosphorylation, N-Myc protein phosphorylated at S62 and N-Myc protein expression; that *lncNB1* knockdown with DOX reduced DEPDC1B expression, ERK protein phosphorylation, N-Myc protein phosphorylated at S62 and N-Myc protein expression; and that forced DEPDC1B over-expression largely blocked *lncNB1* shRNA-mediated reduction in ERK protein phosphorylation, N-Myc protein phosphorylation at S62 and N-Myc protein expression (new Supplementary Fig. 3e). Taken together, our data suggest that *lncNB1* induces ERK protein phosphorylation, N-Myc protein phosphorylation at S62 and N-Myc protein expression by increasing DEPDC1B expression.

Changes:

- We have added the new Alamar blue assay data and the new flow cytometry analysis of apoptosis data from neuroblastoma cells 48 hours after transfection with control siRNA or DEPDC1B siRNAs as the new Supplementary Fig. 4d,e.
- We have added the new rescue experiment immunoblot data as the new Supplementary Fig. 3e.
- We have revised the following sentence from the 2nd line to the 9th line on page 8 in the "Results" section: ".....In addition, transfection with a *lncNB1* or DEPDC1B over-expression construct in BE(2)-C and Kelly cells up-regulated DEPDC1B protein expression, ERK protein phosphorylation as well as N-Myc protein expression (Supplementary Fig. 3d). DOX-inducible *lncNB1* shRNA BE(2)-C and Kelly cells were then transfected with an empty vector or DEPDC1B expression construct and treated with vehicle control or DOX for 48 hours. Immunoblot analysis showed that forced DEPDC1B over-expression largely reversed *lncNB1* shRNA-mediated ERK protein de-phosphorylation, N-Myc protein de-phosphorylation at S62 and N-Myc protein reduction (Supplementary Fig. 3e)....."
- We have added the following sentences as the first paragraph on page 9 in the "Results" section: "To exclude cell growth inhibition or cell death as a contributing factor in the regulation of N-Myc expression, we transfected BE(2)-C and Kelly cells with control siRNA,

lncNB1 siRNAs or DEPDC1B siRNAs for 48 hours, and treated DOX-inducible control shRNA and lncNB1 shRNA BE(2)-C and Kelly cells with vehicle control or DOX for 48 hours. Alamar blue assays and flow cytometry analyses of Annexin V positively stained cells showed that knocking down lncNB1 or DEPDC1B expression for 48 hours was too early to have an effect on neuroblastoma cell proliferation or survival (Supplementary Fig. 4a-e).....”

Question 4. The authors claim from the Kaplan-Meier survival analyses that lncNB1 predicts a poor prognosis of neuroblastoma patients. This is not the case, since expression of lncNB1 is higher in INSS stage 4 neuroblastoma tumors. INSS stage 4 tumors have a very poor prognosis. Neuroblastoma has many INSS stages with a good prognosis (i.e. INSS stage 1, 2 and 4S), which are all included in these analyses. By definition, in a survival analysis that includes neuroblastoma tumors from INSS stage 1, 2, 3, 4 and 4s, a gene that is higher expressed in INSS stage 4 will associate with a poor prognosis. Limiting the analysis to INSS stage 4 neuroblastoma only, the relation between higher expression of lncNB1 and patient prognosis is lost. The author’s conclusion that high levels of lncNB1 predict poor prognosis is therefore of very limited value, since it is not independent from other variables used in the clinical staging of neuroblastoma (e.g. INSS).

Responses:

We agree with the Reviewer that patients with INSS stage 4 neuroblastoma tumors have poorer prognoses than patients with INSS stages 1, 2 and 4S neuroblastoma tumors.

In the original manuscript, using the publicly available SEQC-RPM-seqcnb1 human neuroblastoma tissue RNA sequencing-patient prognosis dataset downloaded from the R2 platform (<http://r2.amc.nl>), we demonstrated that high levels of lncNB1, its partner RPL35 and its targets E2F1 and DEPDC1B expression in 493 human neuroblastoma tissues correlated with poor prognosis in the total population of 493 patients (Fig. 8e-h).

To address the Reviewer’s question, we have analysed the prognostic value of lncNB1 gene expression in stage 4 human neuroblastoma tumor tissues alone, using the same SEQC-RPM-seqcnb1 human neuroblastoma tissue RNA sequencing-patient prognosis dataset. There were 181 INSS stage 4 neuroblastoma samples in the dataset.

Using the median lncNB1 RNA expression as the cut-off point, Kaplan-Meier survival analysis showed that high levels of lncNB1 expression in 181 stage 4 human neuroblastoma tissues were associated with poor patient prognosis in the 181 patients (new Supplementary Fig. 9b). Consistent with this datum, using the median DEPDC1B, E2F1 and RPL35 mRNA expression as the cut-off point, Kaplan-Meier survival analysis showed that high levels of DEPDC1B, E2F1 or RPL35 expression in the 181 stage 4 human neuroblastoma tissues were all associated with poor prognosis in the 181 patients (new Supplementary Fig. 9c-e). Taken together, the data suggest that high levels of lncNB1, DEPDC1B, RPL35 and E2F1 expression in tumor tissues predict poor prognosis in neuroblastoma patients, independent of advanced disease stage.

Changes:

- We have added the new data from Kaplan-Meier survival analysis of lncNB1, DEPDC1B, E2F1 and RPL35 expression in stage 4 human neuroblastoma tumor tissues as the new Supplementary Fig. 9b-e.
- We have added the following sentence from the 8th line to the 10th line on page 14 in the “**Results**” section: “.....In addition, high levels of lncNB1, DEPDC1B, RPL35 or E2F1 RNA

expression in 181 stage 4 neuroblastoma tissues of the SEQC-RPM-seqcnb1 dataset were also associated with poor patient prognosis (Supplementary Fig. 9b-e).

Thank you very much.

Kind regards.

Yours sincerely,

Tao Liu

REVIEWERS' COMMENTS:

Reviewer #1 (Remarks to the Author):

The manuscript is well improved and the authors made several changes according to the comments.

The manuscript is of great interest for the readers of the journal.

Reviewer #2 (Remarks to the Author):

This resubmitted paper improves after careful revision by the authors. However, there are a few residual concerns that need to be fixed.

In response to my previous question #1, the authors state that they can '...safely conclude that lncNB1 is over-expressed in MYCN gene-amplified human neuroblastoma cell lines and tissues, and that lncNB1 over-expression correlates with MYCN gene amplification and overexpression'.

This is unfortunately not the case. lncNB1 is overexpressed in a subset of cell lines with amplification of MYCN. This is evident from the data presented in figure 1e and figure 8a. I suggest to adapt the formulation of row 118 and state that '...lncNB1 was expressed at a significantly higher level in a subset of human neuroblastoma tumors with amplification of MYCN...'.

Furthermore, to give the reader a complete picture of lncNB1 expression in cell lines with or without amplification of MYCN (figure 1f), the expression data of lncNB1 should be visualized as box-dot plots that include the real expression values for individual cell lines. This will present the true variation within cell line compendium, which is absent in the current bar plot representation.

In response to my previous question #2, the authors have performed a substantial amount of experiments to show that there is no phenotypic affect on the cell lines at T=48 hours after transfection. This now justifies their conclusion that knockdown of lncNB1 regulates the selected target genes and that this is a true regulatory relationship, independent of cell cycle and apoptosis. I have no further comments on this.

In response to my previous question #3, the authors have performed a rescue-experiment with DEPDC1B to rescue the regulatory effects on pERK and MYCN after lncNB1 silencing. This experiment now shows that that lncNB1, pERK and MYCN form a true regulatory cascade. I have no further comments on this.

In response to my previous question #4, the authors have performed additional kaplan-meier survival analyses on the selected subset of INSS stage 4 neuroblastoma using the median lncNB1 expression as a cut-off. Unfortunately, this information is in the Supplementary figures. I suggest to combine the survival curves of all INSS stages combined together with the analyses of INSS stage 4 tumors in figure 8. Furthermore, the analyses of INSS stage 4 tumours should also use the 1st-2nd-3rd-4th quartiles as used for all the INSS stage combined.

Additional analysis of an independent neuroblastoma tumor dataset may strongly increase the value of the survival analysis, which I would recommend if possible.

Tao Liu, BMed, MMed, PhD
Group Leader, Histone Modification Group
Children's Cancer Institute Australia
Lowy Cancer Research Centre, UNSW
Randwick, Sydney, NSW 2031
Australia
Ph: 61 2 9385 1935
E-mail: tliu@unsw.edu.au

September 19, 2019

Reviewers & Editors
Nature Communications

Re: Submission of the 2nd revision manuscript NCOMMS-18-34609A entitled "*The long noncoding RNA lncNB1 promotes tumorigenesis by interacting with ribosomal protein RPL35*"

Dear Reviewers and Editors,

Thank you very much for reviewing the manuscript. We appreciate the favourable evaluations and comments. We have revised the manuscript to address the points raised, as detailed below in "Responses to Reviewers" and as indicated in the manuscript files.

We have also revised the manuscript to comply with journal formatting requirements, as indicated in the manuscript files.

RESPONSES TO REVIEWERS

Reviewer #1:

The manuscript is well improved and the authors made several changes according to the comments.

The manuscript is of great interest for the readers of the journal.

Responses:

Thank you. No revisions are required.

Reviewer #2

Question 1. In response to my previous question #1, the authors state that they can '...safely conclude that lncNB1 is over-expressed in MYCN gene-amplified human neuroblastoma cell lines and tissues, and that lncNB1 over-expression correlates with MYCN gene amplification and overexpression'.

This is unfortunately not the case. lncNB1 is overexpressed in a subset of cell lines with amplification of MYCN. This is evident from the data presented in figure 1e and figure 8a. I suggest to adapt the formulation of row 118 and state that '...lncNB1 was expressed at a

significantly higher level in a subset of human neuroblastoma tumors with amplification of MYCN...'.
'.

Furthermore, to give the reader a complete picture of lncNB1 expression in cell lines with or without amplification of MYCN (figure 1f), the expression data of lncNB1 should be visualized as box-dot plots that include the real expression values for individual cell lines. This will present the true variation within cell line compendium, which is absent in the current bar plot representation.

Responses and Changes:

- As suggested by the Reviewer, we have revised the sentence from the 8th to the 10th line on page 6 (row 118 in the first revision manuscript) to the following: “.....lncNB1 was expressed at a significantly higher level in a subset of human neuroblastoma tumors with *MYCN* amplification (Fig. 1f).....”
- As suggested by the Reviewer, we have revised the lncNB1 expression data in Figure 1f from bar plot to box-dot plot so as to include the real lncNB1 expression values for individual cell lines.

Question 2. In response to my previous question #4, the authors have performed additional Kaplan-Meier survival analyses on the selected subset of INSS stage 4 neuroblastoma using the median lncNB1 expression as a cut-off. Unfortunately, this information is in the Supplementary figures. I suggest to combine the survival curves of all INSS stages combined together with the analyses of INSS stage 4 tumors in figure 8. Furthermore, the analyses of INSS stage 4 tumours should also use the 1st-2nd-3rd-4th quartiles as used for all the INSS stage combined.

Additional analysis of an independent neuroblastoma tumor dataset may strongly increase the value of the survival analysis, which I would recommend if possible.

Responses:

In the 1st revision manuscript, using the 1st-2nd-3rd-4th quartiles of RNA expression as the cut-off points, our Kaplan-Meier survival analysis showed that high levels of lncNB1, DEPDC1B, RPL35 and E2F1 RNA expression in the 493 neuroblastoma tissues of the human neuroblastoma tissue RNA sequencing-patient prognosis SEQC-RPM-seqcnb1 dataset were associated with poor patient prognosis (original Fig. 8e-h). In addition, in the 1st revision manuscript, using the median of RNA expression as the cut-off points, our Kaplan-Meier survival analysis showed that high levels of lncNB1, DEPDC1B, RPL35 or E2F1 RNA expression in 181 stage 4 neuroblastoma tissues of the SEQC-RPM-seqcnb1 dataset were also associated with poor patient prognosis (original Supplementary Fig. 9b-e, new Fig. 8i-l).

For the total cohort of 493 neuroblastoma samples in the SEQC-RPM-seqcnb1 dataset, it is easy to perform Kaplan-Meier survival analysis using the 1st-2nd-3rd-4th quartiles of RNA expression as the cut-off points, because there were 123 tumor samples in each of the 4 quartiles. However, for the 181 stage 4 neuroblastoma samples, it is difficult to achieve statistical significance when performing Kaplan-Meier survival analysis using the 1st-2nd-3rd-4th quartiles of RNA expression as the cut-off points, because there were only 45 tumor samples in each of the 4 quartiles. Due to the small sample size of each of the 4 quartiles of the stage 4 tumors, Kaplan-Meier analysis showed high levels of lncNB1 expression in the higher quartile tumors correlated with poor patient prognosis, but the p value was not statistically significant.

To make the Kaplan-Meier survival analysis data from the total cohort of 493 samples and from the sub-cohort of 181 INSS stage 4 samples consistent, we have performed Kaplan-Meier survival analysis of the total cohort of 493 samples of the SEQC-RPM-seqcnb1 dataset, using the median levels of lncNB1, DEPDC1B, RPL35 and E2F1 RNA expression as the cut-off points. The Kaplan-Meier survival analysis showed that high levels of lncNB1, DEPDC1B, RPL35 and E2F1 RNA expression in the total cohort of 493 neuroblastoma tumors also correlated with poor patient prognosis (revised Fig. 8e-h). As suggested by the Reviewer, we have now combined the Kaplan-Meier survival analysis data of the total cohort of 493 samples (of all INSS stages) (Fig. 8e-h) and Kaplan-Meier survival analysis data of the 181 INSS stage 4 samples (Fig. 8 i-l) in Figure 8.

To address the second part of the Reviewer's question, we have examined lncNB1, DEPDC1B, RPL35 and E2F1 RNA expression in the publicly available, much smaller but very well-recognized, microarray gene expression-patient prognosis Versteeg dataset. Kaplan-Meier survival analysis showed that high levels of lncNB1, DEPDC1B, RPL35 and E2F1 RNA expression in the 88 human neuroblastoma tissues also correlated with poor patient prognosis (new Supplementary Fig. 9b-e). Due to the small size of the Versteeg dataset (88 samples in total including only 40 stage 4 tumor samples), it is not possible to perform Kaplan-Meier survival analysis of gene expression in 40 stage 4 tumors of the Versteeg dataset separately.

Changes:

- We have moved the Kaplan-Meier survival analysis data on the prognostic values of lncNB1, DEPDC1B, RPL35 and E2F1 RNA expression in the 181 stage 4 human neuroblastoma tissues of the SEQC-RPM-seqcnb1 dataset, to Figure 8 as the new Fig. 8i-l. We have also revised the Kaplan-Meier survival analysis data on the prognostic values of lncNB1, DEPDC1B, RPL35 and E2F1 RNA expression in the total cohort of 493 human neuroblastoma tissues of the SEQC-RPM-seqcnb1 dataset, from using the 1st-2nd-3rd-4th quartiles of RNA expression as the cut-off points to using the median of RNA expression as the cut-off points (Fig. 8e-h).
- We have added the Kaplan-Meier survival analysis data on the prognostic values of lncNB1, DEPDC1B, RPL35 and E2F1 RNA expression in the 88 human neuroblastoma tissues of the Versteeg dataset as the new Supplementary Fig. 9b-e.
- We have revised the last paragraph on page 13 to the first paragraph on page 14 in the “**Results**” section to the following:

Using the median level of RNA expression as the cut-off point, Kaplan-Meier survival analysis showed that high levels of lncNB1, DEPDC1B, RPL35 and E2F1 RNA expression in the 493 neuroblastoma tissues of the SEQC-RPM-seqcnb1 dataset were associated with poor patient prognosis (Fig. 8e-h). In addition, high levels of lncNB1, DEPDC1B, RPL35 or E2F1 RNA expression in 181 stage 4 neuroblastoma tissues of the SEQC-RPM-seqcnb1 dataset were also associated with poor patient prognosis (Fig. 8i-l). Consistent with these data, high levels of lncNB1, DEPDC1B, RPL35 or E2F1 RNA expression in 88 neuroblastoma tissues of the much smaller publicly available microarray gene expression-patient prognosis Versteeg dataset, which was also downloaded from the R2 platform, were also associated with poor patient prognosis (Supplementary Fig. 9b-e).

Thank you very much for reviewing the manuscript.

Kind regards.

Yours sincerely,

Tao Liu